# NMDAR-mediated modulation of gap junction circuit regulates olfactory learning in *C. elegans*

Myung-Kyu Choi [1,2], He Liu[1,2], Taihong Wu [1,2], Wenxing Yang[1,2] & Yun Zhang [1,2✉]

Modulation of gap junction-mediated electrical synapses is a common form of neural plasticity. However, the behavioral consequence of the modulation and the underlying molecular cellular mechanisms are not understood. Here, using a *C. elegans* circuit of interneurons that are connected by gap junctions, we show that modulation of the gap junctions facilitates olfactory learning. Learning experience weakens the gap junctions and induces a repulsive sensory response to the training odorants, which together decouple the responses of the interneurons to the training odorants to generate learned olfactory behavior. The weakening of the gap junctions results from downregulation of the abundance of a gap junction molecule, which is regulated by cell-autonomous function of the worm homologs of a NMDAR subunit and CaMKII. Thus, our findings identify the function of a gap junction modulation in an in vivo model of learning and a conserved regulatory pathway underlying the modulation.

[1] Department of Organismic and Evolutionary Biology, Harvard University, Cambridge, MA 02138, USA. [2] Center for Brain Science, Harvard University, Cambridge, MA 02138, USA. ✉email: yzhang@oeb.harvard.edu

The function of a neural circuit depends on the dynamic neuronal activities that are regulated by the information flows through electrical and chemical synapses. While various psychiatric and neurodegenerative diseases, including epilepsy and autism, are associated with dysfunctions of neural circuits, the genes and molecules that regulate how neural circuits process signals are not well understood[1–3]. On the other hand, while the physiological properties of various brain circuits are well characterized, their function in regulating animal behavior has not been fully defined, preventing us from connecting genes, neuronal activity and circuit function with behavior.

Electrical synapses, which are mediated by gap junctions and regulate the flow of ions and small molecules between connected cells, are important for neuronal activities[4,5]. Electrical synapses are widely found in the central nervous system, including the hippocampus, and are often located close to the presynaptic or postsynaptic sites or in the same dendritic area as chemical synapses[6,7]. The close localizations facilitate functional interactions between the chemical and the electrical synapses. Similar to the chemical transmission that exhibits synaptic plasticity, electrical synapses are also modulated by neuronal activities. For example, the lateral dendrite of the Mauthner cells in the goldfish auditory system generates both chemical synapse- and electrical synapse-mediated membrane potentials in response to pre-synaptic stimulation. High-frequency stimulation to the input neurons potentiates both the chemical and the electrical synapses in a way that depends on the NMDA-type glutamate receptors (NMDAR)[8,9]. Similarly, in the mammalian inferior olive, stimulating the NMDAR-mediated synapses can either strengthen or weaken the coupling of gap junction-connected neurons depending on the pattern of the stimulation[10,11]. In addition, similar to the metabotropic glutamate receptor (mGluR)-mediated long-term depression, activating the mGluRs in the thalamic reticular nucleus (TRN) weakens the electrical synapses among the TRN neurons[12]. These and several other studies[13,14] demonstrate activity-dependent potentiation or depression of electrical synapses and raise important intriguing questions. For example, are gap junctions modulated during learning? How does the modulation of gap junctions alter the activities of connected neurons? What molecular and cellular events underlie the modulation? Does modulation of gap junctions lead to behavioral changes? Here, we address these questions using *Caenorhabditis elegans*, taking advantage of the wiring diagram of its small nervous system (302 neurons)[15] and the knowledge of the genes encoding the synaptic or gap junction components in many of the neurons[16–20]. This system allows us to examine in vivo the function of conserved molecules in regulating the activity and connectivity of neural circuits with genetics and imaging tools and to mechanistically dissect the molecular and cellular bases for the dynamics of circuit activity and their function in behavior.

In the *C. elegans* sensorimotor circuit, several pairs of chemosensory neurons detect bacterially generated volatiles to regulate odorant-guided locomotion[21–25]. Among these neurons, AWC sense attractive odorants and send chemical synapses to the interneurons AIB and AVA, both of which are connected with the interneurons RIM with chemical and electrical synapses[15] (Fig. 1a, b). Exposure to attractive odorants, such as isoamyl alcohol (IAA), reliably suppresses AWC activity and generates variable but correlated suppression for AIB, AVA, and RIM. Manipulating the activity of either RIM or AIB alters the odorant-evoked responses of the other two interneurons, demonstrating that these interneurons act together to encode sensory information[26]. RIM and AVA are connected with motor neurons to regulate reorienting movements, such as reversals and turns. The activities of RIM, AIB, and AVA are all correlated with reversals and activating any of these three neurons induces reorienting

movements. Thus, simultaneously suppressing AIB, AVA, and RIM by attractive odorants, such as IAA, reduces reversals and facilitates forward movement towards IAA[24,26–31]. Here, in this study we employ the circuit of these interneurons to study the behavioral consequence and the regulatory mechanisms of modulating gap junctions.

We previously showed that *C. elegans* learned to avoid the smell of pathogenic bacteria, such as *Pseudomonas aeruginosa* strain PA14[23,32]. Here, we show that exposure to PA14-generated odorants suppresses the activities of AIB, AVA, and RIM and training with PA14 decouples these responses. The decoupling depends on the weakening of the gap junctions of RIM and a training-induced repulsive sensory response to PA14. Blocking training-induced weakening of RIM-gap junctions abolishes training-dependent decoupling and disrupts learning. Furthermore, we show that training-dependent weakening of RIM-gap junctions is regulated by the worm homolog of a mammalian NMDAR subunit, NMR-1, and its downstream effector CaMKII in RIM, which reduce the abundance of the gap junction molecule INX-4 after training. Together, our results demonstrate the function of modulating gap junctions in an in vivo model of learning and identify the NMDAR- and CaMKII-mediated cell-autonomous downregulation of gap junction molecules as the mechanism underlying the modulation.

## Results

**Olfactory learning modulates RIM, AIB, and AVA.** Adult *C. elegans* learns to reduce its preference for the smell of pathogenic bacteria, such as the *Pseudomonas aeruginosa* strain PA14[33], after ingesting the pathogen for several hours[23,32,34,35]. To quantify olfactory learning of PA14 in adult animals, we used a previously established automated assay to measure olfactory preference in swimming worms that were stimulated by air streams of tested odorants[23] (Supplementary Fig. 1 and "Methods"). When swimming, a worm continuously and slightly bends its body, which is occasionally disrupted by large bends that have a shape of the Greek letter Ω. Because Ω-bends are followed by reorienting movements, such as reversals, the rate of Ω-bends inversely correlates with the preference of the worm for the tested odorant[23,36]. Thus, we measured olfactory preference in worms by quantifying the rate of Ω-bends.

We grew a cohort of animals under the standard condition[37] and trained some adults by feeding them on a lawn of PA14 for 4–6 hours while feeding the rest of the worms on a lawn of the standard food, *Escherichia coli* strain OP50, as naive controls (Supplementary Fig. 1 and "Methods"). Previously, by testing the preference between the odorants of PA14 and the odorants of OP50 in worms, we found that training with PA14 shifted their preference towards OP50 and away from PA14[23,32]. Thus, we first asked whether training altered the preference for PA14 as well as for OP50. Because worms, even after training with the pathogenic bacterium PA14, prefer food odorants, including the odorants of PA14, over non-food odorants[23], we could not use non-food odorants as controls to test training-induced changes in their preference for food odorants. Previous studies have shown that animals classify odorants based on concentration and that a fourfold dilution is sufficient for a robust classification[38]. Thus, we diluted the culture of PA14 (referred to as PA14) by fourfold and used the diluted PA14 as a control (referred to as PA-control) to examine whether training with PA14 reduced the preference for the odorants of PA14. We found that naive animals preferred the smell of PA14 to the smell of PA-control and that training with PA14 strongly reduced this preference (Fig. 1c and "Methods"). In contrast, when tested with the culture of OP50 (referred to as OP50) versus the fourfold diluted culture of OP50

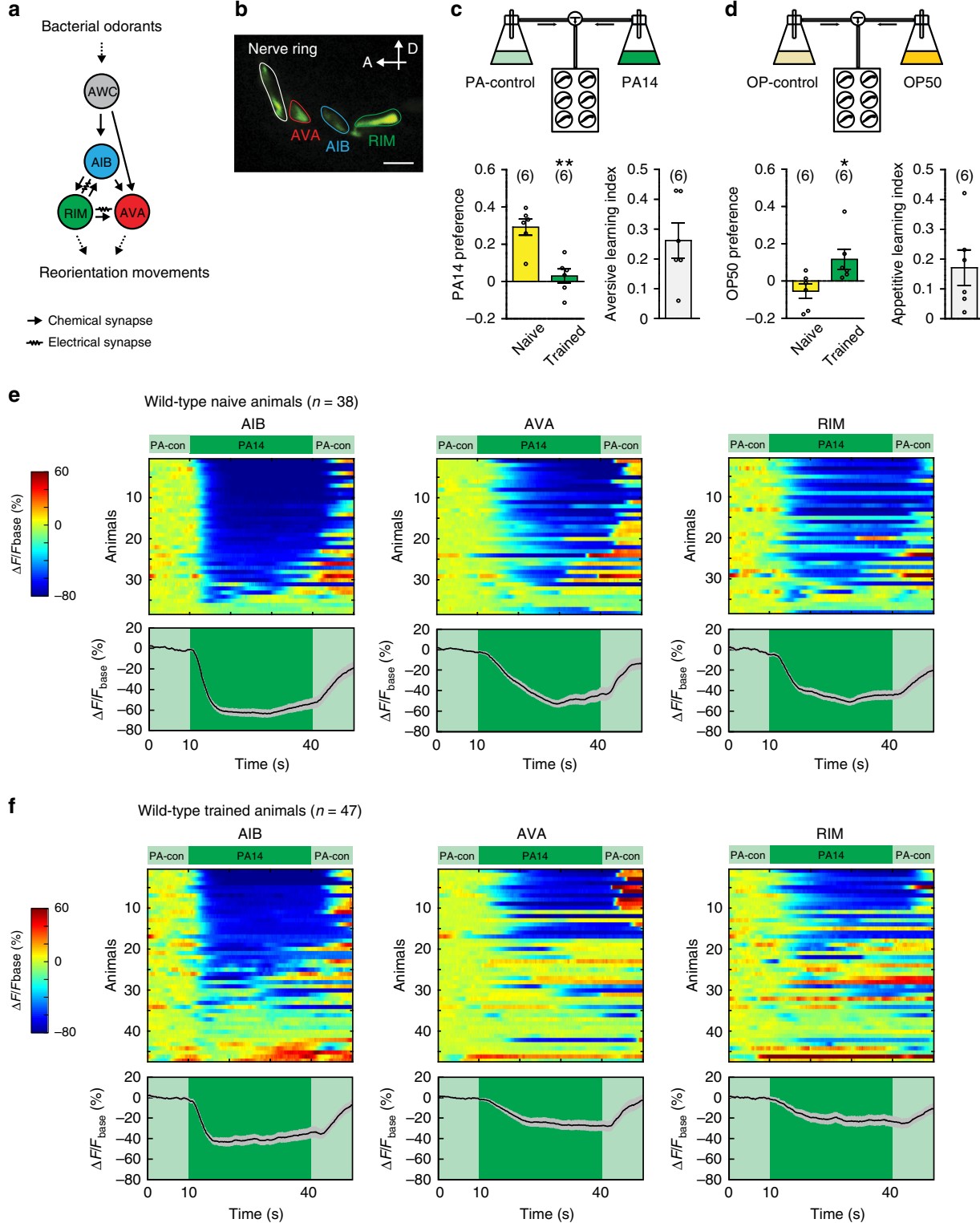

**e** Wild-type naive animals (n = 38)

**f** Wild-type trained animals (n = 47)

(referred to as OP-control) naive animals did not show a strong preference, but training with PA14 increased the preference for OP50 (Fig. 1d and "Methods"). These results demonstrate that training with PA14 generates aversive learning of PA14 and appetitive learning of OP50.

Previous studies showed that AIB, AVA, and RIM interneurons regulated the learning of PA14 in adult animals[23,31,32,34]. To characterize the function of these neurons in learning, we recorded intracellular calcium transients in transgenic animals

that expressed GCaMP3[39] in these neurons[26] (Fig. 1b). We used a microfluidic device[27] to stimulate the animals with the supernatants of a culture of PA14 and PA-control, and recorded the GCaMP3 signals of AIB, AVA, and RIM ("Methods"). We found that switching from PA-control to PA14 suppressed the intensity of the GCaMP3 signals in all three pairs of the neurons in naive animals (Fig. 1e). This finding is consistent with the behavioral preference for PA14 in naive worms, because it has been shown that attractive odorants suppress the activities of AIB, AVA, and

**Fig. 1 Olfactory learning modulates sensory-evoked responses in AIB, AVA, and RIM. a** A schematic diagram showing the wiring of interneurons AIB, AVA, RIM, and sensory neurons AWC. **b** A sample image of GCaMP3 expression in AIB, AVA, and RIM. Similar expression patterns are observed for all animals imaged. Scale bar, 5 μm; A anterior; D dorsal. **c**, **d** Schematic diagrams showing the droplet assay (top) and olfactory training effects (bottom). Naive animals prefer the smell of PA14 to PA-control, and training with PA14 decreases the preference (**c** bottom, $p = 0.0011$); naive animals do not have a clear preference between the smell of OP50 and OP-control, and training with PA14 increases the preference for OP50 (**d** bottom, $p = 0.030$). The choice index and learning index are defined in Methods. Naive and trained animals are compared using two-sided Welch's *t*-test, mean ± s.e.m., parentheses contain the numbers of assays measured over 5 (**c**) or 4 (**d**) independent experiments, circles indicate individual data points. **e**, **f** GCaMP3 signals of AIB, AVA, and RIM evoked by switches between PA14 and PA-control (PA-con) in individual animals (top) and mean values (bottom). The GCaMP3 signals of RIM, AIB and AVA are simultaneously recorded in every animal and arranged based on the response of AIB. ΔF = F−Fbase and Fbase is average GCaMP3 signal in the 10 s window before PA14 stimulation ("Methods"). Spectrums indicate the ranges of ΔF/Fbase (%) and ΔF/Fbase (%) are outside of indicated ranges for some frames. Solid traces and shades in the bottom panels respectively denote mean values and s.e.m. Parentheses contain the numbers of animals measured over nine independent experiments. For **c** and **d** asterisks indicate significant difference, \*\**p* < 0.01, \**p* < 0.05. Source data are provided as a Source Data file.

RIM to suppress reorienting reversals and turns[24,26–31,34]. After training, the PA14-evoked GCaMP3 responses in these neurons became weaker and more variable (Fig. 1f). To characterize training-dependent changes in PA14-evoked responses in AIB, AVA, and RIM, we quantified several parameters. First, we found that the average amplitude of PA14-evoked responses was significantly smaller in all three pairs of the interneurons after training (Fig. 2a, Supplementary Fig. 2 and "Methods"). The difference in the average response amplitude started to manifest at 10 s after the onset of PA14 stimulation and continued throughout the 30 s exposure to PA14 (Fig. 2a, Supplementary Figs. 3a and 4a). Second, we found that the duration of PA14-evoked responses in these neurons also significantly decreased after training (Fig. 2b and Supplementary Fig. 2). We analyzed response duration using three different amplitude thresholds, i.e., 10%, 30%, and 50% decrease in GCaMP3 signal (Fig. 2b, Supplementary Figs. 5a and 6a), and reached the same conclusion. Third, because these interneurons regulate each other[26] and the patterns of their PA14-evoked calcium responses appear more variable in trained animals than in naive animals (Fig. 1e, f), we asked whether training modulated the correlation of their activities. First, we analyzed the pairwise cross–correlation of the GCaMP3 signals of AIB, AVA, and RIM over time using a 10 s sliding window (Supplementary Fig. 7a). In naive animals, the correlation coefficients for AIB-AVA, AIB-RIM, and AVA-RIM start to increase when the sliding window enters the time window of PA14-exposure and peak for several seconds before they gradually decrease (Supplementary Fig. 7b), indicating that the correlation of the neuronal activities increases in response to PA14 stimulation. Training with PA14 decreases the correlation of the neuronal activities (Supplementary Fig. 7b). To quantify this effect, we measured the correlation coefficients of the GCaMP3 signals during the 30 s exposure to PA14 and found significant decreases in trained animals compared with naive animals (Fig. 2c and "Methods"). Together, these results indicate that the aversive training not only reduces the amplitude and the duration of PA14-evoked responses in these interneurons but also decouples the responses. In contrast, none of these three parameters of OP50-evoked responses in these neurons changed after training (Fig. 2d–f and Supplementary Figs. 3b, 4b, 5b, 6b and 8), demonstrating specific training effects on PA14-evoked responses.

To understand how the training-dependent changes in PA14-evoked activities of AIB, AVA, and RIM alter the activity ensemble of these neurons, we quantified the time derivatives of the GCaMP3 signals. A positive or a negative time derivative respectively indicates an increase or a decrease in the intracellular calcium level for the time window measured and represents whether the neurons is becoming more or less active during the time[40] ("Methods"). In total, there are eight different patterns of

the activity ensemble with each representing a unique combination of the activities of these neurons (Fig. 2g). We determined the portion of time when each of the eight possible activity patterns was displayed by the circuit in four different time windows in every animal during PA14 exposure (i.e., 0–15 s, 5–15 s, 0–20 s, and 0–30 s of PA14 exposure) and compared the mean values in naive and trained animals. We found that the pattern in which AIB, AVA, and RIM were all inhibited (AIB↓ AVA↓ RIM↓) was predominant in PA14-evoked calcium responses in naive animals and the percentage of time when the circuit displayed this pattern significantly decreased in trained animals (Fig. 2g and Supplementary Fig. 9). This decrease was consistently accompanied by significant increases in another two states (AIB↓ AVA↑ RIM↓, AIB↑ AVA↑ RIM↓, Fig. 2g and Supplementary Fig. 9). In contrast, the activity patterns evoked by PA-control or by OP50 were not significantly altered by training (Fig. 2g and Supplementary Fig. 9). Because activating any of AIB, AVA, and RIM promotes reversals[24,26–31], the decreased percentage of the activity pattern for which AIB, AVA, and RIM were all inhibited by PA14 in trained animals predicts a higher rate of reorienting movements in response to PA14. As predicted, we found that the turning rate evoked by PA14 was significantly higher in trained animals than in naive animals, but the turning rates evoked by PA-control were comparable (Fig. 2h). Because a higher turning rate indicates a lower preference for the tested stimulus[23,36], these results show that the training-dependent changes in PA14-evoked activity patterns of the interneurons generate a reduced preference for PA14 in trained animals. Because the analysis on activity patterns for four different time windows generated the same conclusion, in the rest of the paper we analyzed the pattern of the circuit activity using the 0–20 s time window after the onset of PA14. Together, these results demonstrate specific training effects on PA14-evoked sensory responses in AIB, AVA, and RIM and suggest the function of these learning-dependent changes in regulating aversive learning of PA14, which is the focus for the rest of this study.

**Decoupling AIB, AVA, and RIM facilitates olfactory learning.** To characterize the function of training-induced activity changes in AIB, AVA, and RIM, we sought the underlying molecular and cellular mechanisms. AIB, AVA, and RIM are connected through chemical synapses and gap junction-mediated electrical synapses[15] (Fig. 3a). Because gap junctions couple neuronal activities, we examined whether the training-dependent changes in AIB, AVA, and RIM, including the decoupling of their activities, resulted from weakening of the gap junctions in these cells. We expressed one mammalian neuronal gap junction molecule connexin Cx36[5,41] in RIM, AIB, and AVA (Fig. 3a) using promoters selectively expressed in these neurons[19,29,31,42]. Previous studies

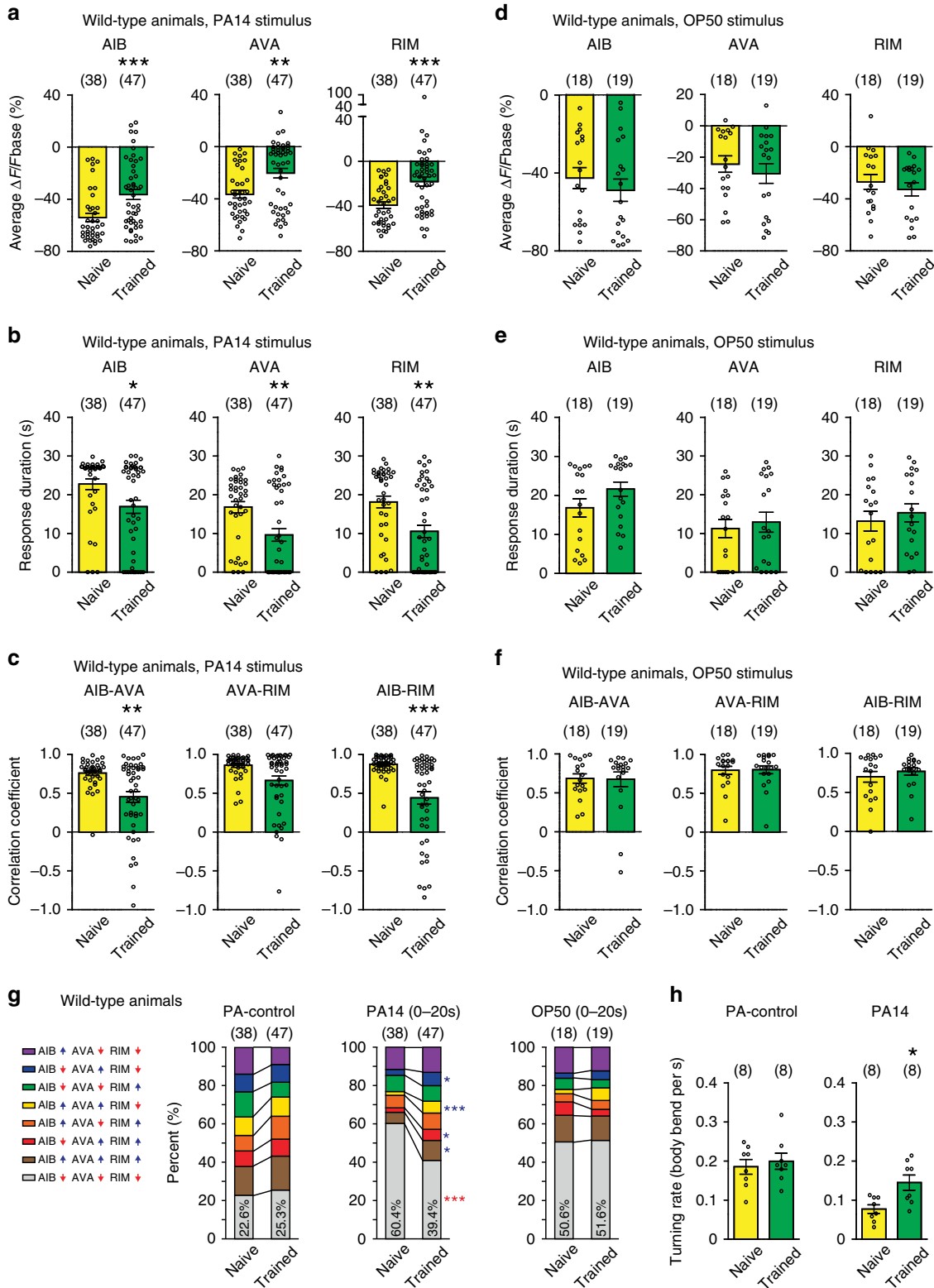

showed that ectopic expression of Cx36 in *C. elegans* neurons that were physically adjacent to each other generated gap junctions that coupled the neurons[43]. We found that expressing Cx36 in AIB, AVA, and RIM abolished the training-induced changes in the amplitude, duration, and correlation coefficients of PA14-evoked GCaMP3 signals in these neurons (Figs. 3b–f and Supplementary Figs. 3c, 4c, 5c, 6c). Consistently, the patterns of PA14-evoked activities in these interneurons were also similar in

naive and trained Cx36-expressing animals (Fig. 3g and Supplementary Fig. 9g). These results indicate that strengthening the gap junction-mediated coupling of AIB, AVA, and RIM antagonizes the training-induced activity changes in these neurons. Interestingly, as a result of blocking training-dependent decoupling of the interneurons, the aversive olfactory learning of PA14 was completely abolished in the transgenic animals expressing Cx36 (Fig. 3h). The loss of learning in the Cx36-expressing animals

**Fig. 2 Olfactory learning modulates PA14-evoked responses in AIB, AVA, and RIM. a–c** The average response amplitude (**a**), response duration (**b**), correlation coefficients (**c**) of PA14-evoked GCaMP3 signals of AIB, AVA, and RIM in naive and trained wild-type animals ("Methods"). **d–f** The average response amplitude (**d**), response duration (**e**), and correlation coefficients (**f**) of OP50-evoked GCaMP3 signals of AIB, AVA, and RIM in naive and trained wild-type animals ("Methods"). For (**a–f**), average response amplitude and correlation coefficients are measured for GCaMP3 signals during 30 s PA14 stimulation or 30 s OP50 stimulation (**a, c, d, f**), and response duration is measured for responses with ΔF/Fbase < −30% (**b, e**). **g** Patterns of GCaMP3 signals in AIB, AVA, and RIM that are evoked by PA14 (0–20 s exposure to PA14), or by PA-control, or by OP50 (0–20 s exposure to OP50) in wild-type animals. Upward pointing or downward pointing arrow following each neuron respectively denotes a positive or a negative value for time derivative of GCaMP3 signal of the neuron ("Methods"). Mean values are presented here, and Supplementary Fig. 9 show mean, s.e.m., individual data points, and *p* values. For (**a–g**), parentheses contain the numbers of animals examined over 9 (PA14 exposure) or 3 (OP50 exposure) independent experiments. **h** Turning rates in response to PA14 and PA-control in naive and trained wild-type animals, parentheses contain the numbers of assays measured over eight independent experiments. For all, naive and trained animals are compared using two-sided Mann–Whitney *U* test (**a–f**) or two-sided Mann–Whitney *U* test with Bonferroni correction (**g**) or two-sided Welch's *t*-test (**h**), asterisks indicate significant difference, \*\*\**p* < 0.001, \*\**p* < 0.01, \**p* < 0.05, mean ± s.e.m., circles indicate individual data points. The *p* values in the following panels, from left to right, are: **a** 0.0005, 0.0019, <0.0001; **b** 0.014, 0.0039, 0.0011; **c** 0.0047, 0.10, <0.0001; **d** 0.40, 0.37, 0.34; **e** 0.54, 0.43, 0.45; **f** 0.66, 0.91, >0.99; **h** 0.62, 0.011. Source data are provided as a Source Data file.

resulted from a significantly decreased PA14-evoked turning rate after training in comparison with non-transgenic animals. In contrast, the expression of Cx36 did not alter the turning rate under the naive condition (Fig. 3i), indicating that ectopically expressing Cx36 modulates PA14-evoked responses specifically in trained animals. To further confirm the learning defect generated by strengthening the gap junctions, we measured olfactory learning in the transgenic animals using another established assay in which a worm crawled towards a drop of supernatant of freshly prepared culture of PA14 on a plate[44]. Consistent with our previous findings, we showed that training adult worms with PA14 for 4–6 h decreased the efficiency of their odorant-guided movements towards PA14, which was measured by a navigation index, and increased the distance traveled to reach PA14, indicating a reduced preference for PA14[44] (Supplementary Fig. 10a). Using this assay, we found that expressing Cx36 disrupted the aversive learning (Supplementary Fig. 10b, c), consistent with the results generated by using the droplet assay. RIM form electrical synapses with AIB and AVA[15]. We found that expressing Cx36 in RIM and AIB was sufficient to disrupt learning, while expressing Cx36 in RIM and AVA had no effect (Fig. 3j, k). Thus, learning requires the decoupling of PA14-evoked responses in AIB, AVA, and RIM, which produces an increased turning rate in response to PA14 and a decreased preference for PA14.

**A sensory response regulates specificity of learning.** We next asked if training modulated RIM, AIB, and AVA, why only PA14-evoked responses in these neurons, but not OP50-evoked responses, were decoupled after training. We hypothesized that the specificity of learning occurred upstream of the circuit. To test this possibility, we examined the pair of nociceptive sensory neurons ASH, because ASH send synapses to all of AIB, AVA, and RIM[15]. ASH respond to repulsive stimuli, such as copper, with increased intracellular calcium transients and activating ASH induces avoidance by generating reversals[45,46]. Using a transgenic line that expressed GCaMP6[47] in ASH, we found that ASH in naive animals did not respond to the stimulation of PA14 (Fig. 4a). However, after training with PA14, ASH became strongly activated by PA14 (Fig. 4a, b), indicating that ASH in PA14-trained animals respond to PA14 as a repulsive cue. This training-dependent modulation of ASH is specific for PA14, because ASH in naive and trained animals similarly respond to OP50 (Fig. 4c, d). In comparison, the main olfactory sensory neurons AWC that sense food odorants upstream of AIB, AVA, and RIM do not show a significant change in their PA14-evoked calcium responses after training (Fig. 4e, f) and continue to respond to PA14 as an attractive cue. Genetically ablating ASH[48]

strongly disrupted aversive learning of PA14, demonstrating the critical role of ASH in learning (Fig. 4g). Together, these results show that the nociceptive sensory neurons ASH provide the repulsive information of PA14 to AIB, AVA, and RIM in trained animals and this sensory information converges with the attractive information of PA14 transmitted by AWC to generate decoupled responses specific to PA14 in trained animals.

**A NMDA receptor in RIM regulates decoupling and learning.** Next, we sought the mechanisms underlying training-induced decoupling. We examined *nmr-1*, because it encodes the *C. elegans* homolog of the NR1 subunit of the mammalian NMDARs[49], which play a critical role in regulating synaptic plasticity in the mammalian brain[50]. Meanwhile, *nmr-1* is expressed in a few *C. elegans* neurons, including RIM[49]. We found that the *nmr-1(ak4)* loss-of-function mutation disrupted the aversive olfactory learning of PA14 in both the droplet assay and the plate assay (Fig. 5a and Supplementary Fig. 10d). In contrast, the *nmr-1* mutant animals were normal in appetitive learning of OP50 (Fig. 5a), indicating that the defect of *nmr-1* mutants are specific in learning of PA14, but not general in neuronal functions. Expressing *nmr-1* in RIM alone using the cell-selective promoter $P_{gcy-13}$[42] fully rescued the defect of the *nmr-1(ak4)* mutants in generating aversive learning (Fig. 5b and Supplementary Fig. 10e, f). These results together indicate that *nmr-1* acts in the RIM interneurons to regulate learning of PA14.

To characterize the function of *nmr-1* in learning, we examined the GCaMP3 signals of AIB, AVA, and RIM. First, we found that stimulating the *nmr-1* mutant animals with PA14 suppressed the activities of AIB, AVA, and RIM, as indicated by the GCaMP3 signals in these neurons. However, training with PA14 no longer reduced the amplitude or the duration of PA14-evoked responses in these neurons in the *nmr-1* mutant animals and it also did not alter the correlation coefficients of the GCaMP3 signals (Figs. 5c–g and Supplementary Figs. 3d, 4d, 5d, 6d). In addition, the PA14-evoked activity patterns of the interneurons were comparable in naive and trained *nmr-1* mutant animals (Fig. 5k and Supplementary Fig. 9h). Remarkably, expressing a wild-type copy of *nmr-1* in RIM rescued the defects in PA14-evoked neuronal responses in AIB, AVA and RIM in the *nmr-1* mutants (Fig. 5h–k and Supplementary Figs. 3e, 4e, 5e, 6e, 9i, and 11), which indicates that NMR-1 acts in RIM to regulate training-dependent changes in the activities of AIB, AVA and RIM to generate learning. The findings showing that the RIM-expressed *nmr-1* rescues the activity of RIM and that of AIB and AVA further demonstrate that AIB, AVA and RIM act as a network to regulate each other. We will refer to AIB, AVA, and RIM as RIM-circuit henceforward.

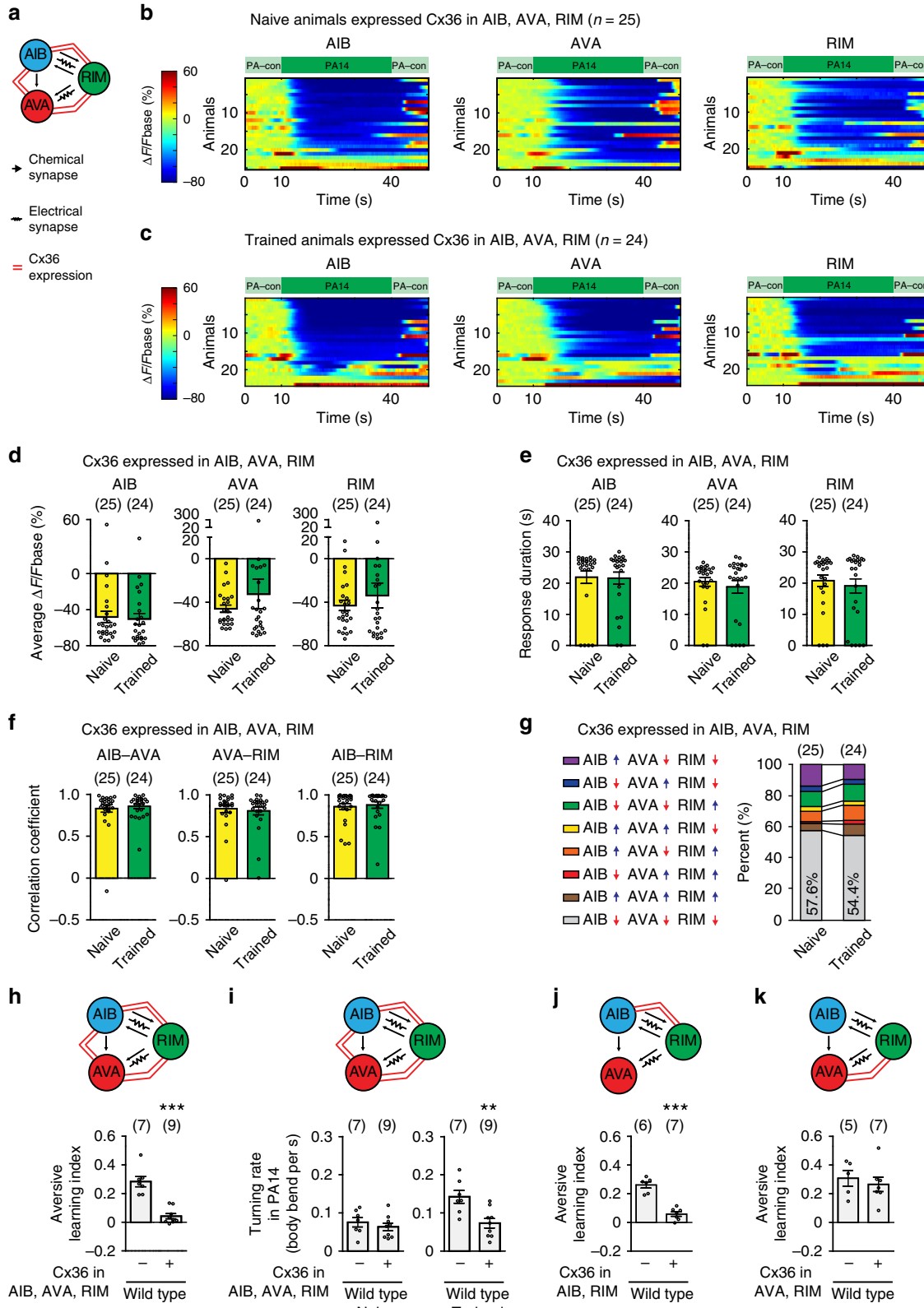

**INX-4 acts downstream of NMR-1 to decouple RIM-circuit.** Next, we asked what blocked the training-dependent decoupling in the *nmr-1* mutant animals. Because strengthening the coupling of RIM-circuit by ectopically expressing the mammalian neuronal connexin Cx36 inhibits the training-induced changes in RIM-circuit (Fig. 3), we hypothesized that misregulation of gap junctions in the *nmr-1* mutants disrupted training-dependent

decoupling. Thus, we examined the gap junction molecules expressed in RIM. Connexins and innexins form gap junctions in chordates and prechordates, respectively. Although innexins and connexins differ in their protein sequences, they form gap junctions that are highly similar in structure[5,51,52]. The *C. elegans* genome encodes 25 innexins that are expressed in neurons or muscles to electrically couple the connected cells[53–55]. *inx-4*

**Fig. 3 Decoupling PA14-evoked responses of RIM-circuit facilitates learning. a** A schematics showing Cx36 ectopic expression. **b, c** GCaMP3 signals of AIB, AVA, and RIM evoked by PA14 and control in transgenic animals expressing Cx36 in AIB, AVA, RIM. GCaMP3 signals of AIB, AVA, RIM are simultaneously recorded in every animal and arranged based on GCaMP3 of AIB. $\Delta F = F - Fbase$, Fbase is the average GCaMP3 signal in 10 s window before PA14 stimulation. Spectrums indicate the ranges of $\Delta F/Fbase$ (%), which are outside of indicated ranges for some frames. **d–g** Average response amplitude (**d**), response duration (**e**), correlation coefficients (**f**), activity patterns (**g**) of PA14-evoked GCaMP3 signals in AIB, AVA, and RIM in naive and trained transgenic animals expressing Cx36 in AIB, AVA, and RIM. Average response amplitude and correlation coefficients are for GCaMP3 signals during 30 s PA14 stimulation (**d, f**), response duration is for response with $\Delta F/Fbase < -30\%$ (**e**). Activity patterns are for the first 20 s PA14 exposure (**g**). In **g**, upward pointing or downward pointing arrow respectively denotes a positive or a negative value for time derivative of GCaMP3. For (**d–g**), naive and trained animals are compared using two-sided Mann–Whitney $U$ test (**d–f**) or two-sided Mann–Whitney $U$ test with Bonferroni correction (**g**), mean ± s.e. m. **d–f** mean (**g**). Supplementary Fig. 9 shows mean ± s.e.m., $p$ values, individual data points for **g**. Parentheses contain the numbers of animals examined over 7 independent experiments (**b–g**), circles indicate individual data points. **h–k** Expressing Cx36 in RIM, AIB and AVA disrupts aversive learning (**h** $p = 0.0002$) by altering turning rate (**i** naive $p = 0.45$, trained $p = 0.0067$); expressing Cx36 in RIM and AIB disrupts learning (**j** $p < 0.001$), but expressing Cx36 in RIM and AVA does not (**k** $p = 0.59$), transgenic animals (+) are compared with non-transgenic controls (−) using two-sided Welch's $t$-test, mean ± s.e.m. Parentheses contain the numbers of assays measured over 5 (**h, i**) or 4 (**j, k**) independent experiments, circles indicate individual data points. For (**d–k**), asterisks indicate significant difference, ***$p < 0.001$, **$p < 0.01$, *$p < 0.05$. Source data are provided as a Source Data file.

encodes an innexin expressed in RIM but not in AIB or AVA[19,20]. We found that removing *inx-4* with a deletion mutation *ok2373* suppressed the defect of *nmr-1(ak4)* mutant animals in generating training-induced decoupling of RIM-circuit. Exposure to PA14 suppressed the GGaMP3 signals in RIM-circuit in naive *nmr-1;inx-4* double mutant animals and training with PA14 significantly reduced the amplitude and the duration of the PA14-evoked GCaMP3 signals (Fig. 6a–d and Supplementary Figs. 3f, 4f, 5f, 6f). The pairwise cross-correlations of PA14-evoked sensory responses of RIM-circuit in the *nmr-1;inx-4* mutants were significantly decreased by training (Fig. 6e). The training-induced activity changes generate a significant decrease in the percentage of time when all three interneurons were inhibited by PA14 (Fig. 6f and Supplementary Fig. 9j). In addition, restoring *inx-4* expression in RIM reversed the suppressing effect of mutating *inx-4* on the *nmr-1* mutant animals (Supplementary Fig. 12). Thus, in *nmr-1* mutant animals the gap junction molecule *inx-4* blocks the training-induced decoupling of PA14-evoked responses in RIM-circuit. These results together with the findings showing that ectopically expressing Cx36 in RIM and AIB blocks the training-induced modulation of RIM-circuit in wild type indicate that NMR-1 acts in RIM to generate training-dependent changes in RIM-circuit by suppressing the INX-4-mediated gap junctions.

**Training downregulates INX-4 via NMR-1 and UNC-43 in RIM.** Next, we addressed how NMR-1 suppresses INX-4 to regulate learning. We first examined a transcriptional reporter that expressed *gfp* with an *inx-4* promoter ($P_{inx-4}::gfp$). The reporter is expressed in several neurons, including RIM[19,20]. We quantified the intensity of the GFP signal in RIM soma in naive and trained animals using confocal microscopy ("Methods") and did not detect any difference (Fig. 7a). Next, we generated a translational fusion $P_{inx-4}::gfp::inx-4$ by fusing the *inx-4* cDNA to the coding sequence of *gfp* and expressed the fusion with the *inx-4* promoter. The translational fusion was functional, because it rescued the hypersensitive response to quinine in the *inx-4* mutant animals[56] (Supplementary Fig. 13). We found that the GFP::INX-4 signal in RIM soma was significantly weaker in trained animals than in naive animals (Fig. 7b, c). Because the *inx-4* transcriptional reporter and the *gfp::inx-4* translational fusion shared the same 5' and 3' regulatory sequences, the training-induced decrease in the GFP::INX-4 signal in RIM resulted from a post-transcriptional regulation, such as decreased protein abundance. Importantly, the training-dependent decrease in GFP::INX-4 was abolished by the *nmr-1* mutation (Fig. 7d, e), indicating that *nmr-1* regulates learning by downregulating the abundance of INX-4 in RIM to decouple the PA14-evoked

responses in RIM-circuit. The nociceptive sensory neuron ASH is one of the glutamatergic neurons that are presynaptic to RIM[15]. We showed that ablating ASH disrupted learning (Fig. 4g). However, we found that genetically ablating ASH did not significantly alter the training-induced downregulation of INX-4 in RIM (Fig. 7f). These results together show that the training-dependent modulation of RIM-gap junctions does not require the function of ASH or the training-induced modulation of ASH.

We then addressed how NMR-1 downregulated INX-4 abundance. In the vertebrate nervous system, CaMKII regulates various forms of NMDAR-dependent synaptic plasticity[57]. *unc-43* encodes the only worm homolog of the primary form of CaMKII in the brain[58,59]. We selectively expressed in RIM a constitutively active form of UNC-43, T286D. The UNC-43(T286D) equivalent form of CaMKII mimics the autophosphorylated CaMKII that is constitutively active independent of calcium and calmodulin[58–62]. We found that expressing UNC-43(T286D) in RIM significantly reduced the signal of GFP::INX-4 in RIM in wild-type animals even without training (Fig. 7g). In contrast, expressing UNC-43 (T286D) in RIM did not alter the expression of the $P_{inx-4}::gfp$ transcriptional reporter (Fig. 7h). These results together show that the activated UNC-43 acts in RIM to reduce the abundance of INX-4. Next, we tested whether the activated UNC-43 acts downstream of NMR-1 to regulate learning by expressing UNC-43(T286D) in the RIM neurons of the *nmr-1* mutant animals. We found that, strikingly, the *nmr-1* mutants that expressed UNC-43 (T286D) in RIM learned to reduce their preference for PA14 similarly as wild-type animals, both in the droplet assay and in the plate assay (Fig. 7i and Supplementary Fig. 10g, h). Together, these results demonstrate that NMR-1 acts in RIM to downregulate INX-4 abundance in trained animals through the activated UNC-43/CaMKII, which weakens the gap junction in RIM-circuit. The RIM-circuit with the weaker gap junctions acts together with the repulsive information of PA14 generated by ASH to produce training-dependent reduction in the preference for PA14 (Supplementary Fig. 14).

**Discussion**
A neural circuit exhibits complex spatial and temporal activity patterns that are generated by the activity of individual neurons and the interactions among them. To better understand the function of a neural circuit, it is essential to address how sensory information is encoded and processed by these high-dimensional signals to produce behavior and how experience modulates the property of the circuit to generate learning. The variable but coupled activities of three gap junction-connected interneurons, i.e., AIB, AVA, and RIM, in *C. elegans* respond to food odorants to direct behavioral preference for the odorants by regulating

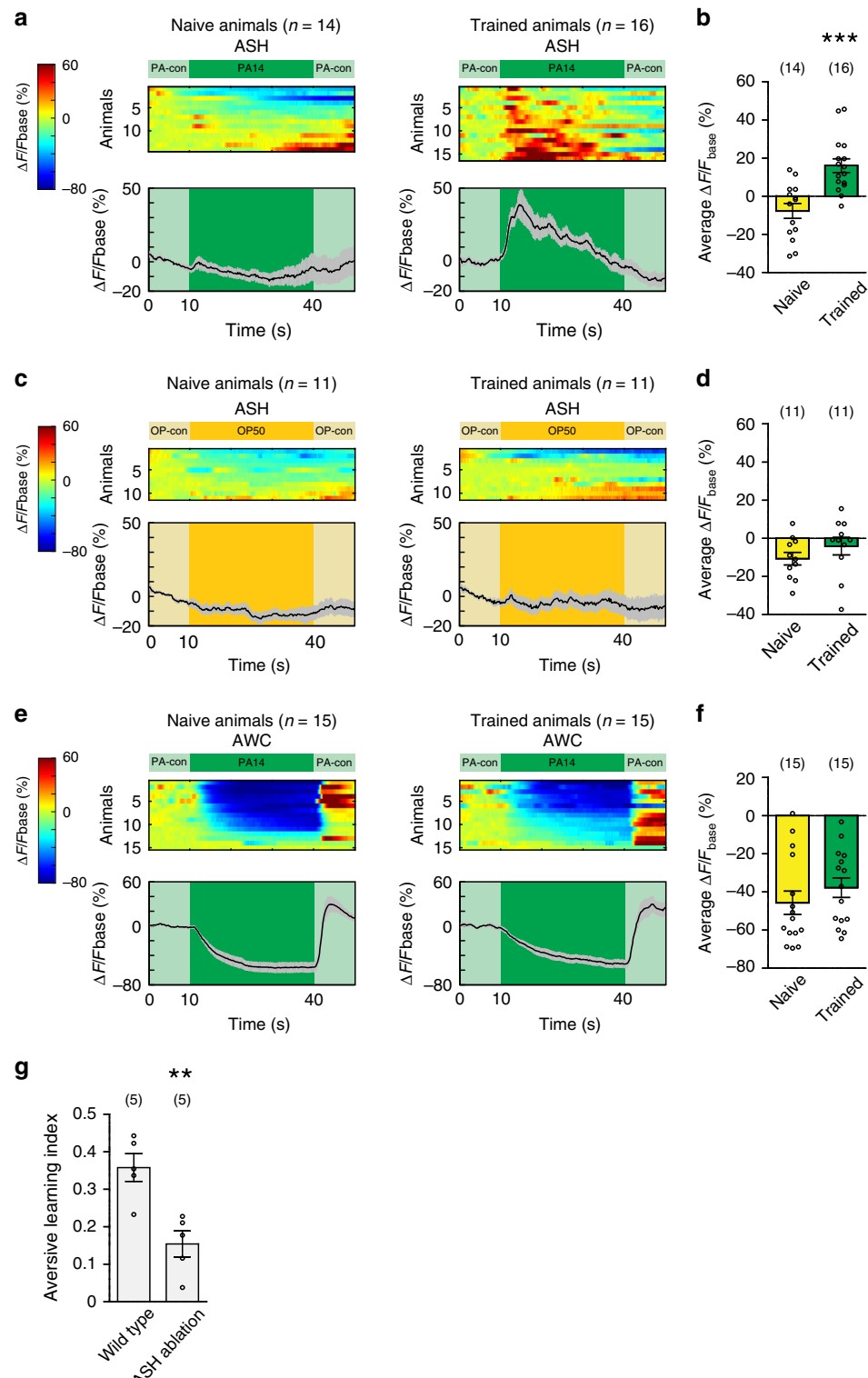

**Fig. 4 Olfactory learning modulates PA14-evoked response of ASH. a–f** GCaMP6 signals of ASH (**a–d**) and AWC (**e, f**) evoked by PA14 and control (**a, b, e, f**) or by OP50 and control (**c, d**) in naive and trained animals. ΔF = F−Fbase and Fbase is the average GCaMP6 signal in the 10 s window before PA14 or OP50 stimulation. Spectrums indicate the ranges of ΔF/Fbase (%) and ΔF/Fbase (%) are outside of indicated ranges for some frames. In **a, c,** and **e,** top panels show GCaMP6 signals of ASH or AWC in individual animals, which are arranged based on response amplitude, and bottom panels show mean values and s.e.m. In **b, d,** and **f,** average response amplitude is measured for GCaMP6 signals during 30-s stimulation of PA14 or OP50. Parentheses contain the numbers of animals examined over 4 (**a, b**) or 2 (**c–f**) independent experiments, circles indicate individual data points, mean ± s.e.m. Naive and trained animals are compared using two-sided Welch's *t*-test (**b** $p = 0.0001$; **f** $p = 0.33$) or two-sided Mann–Whitney *U* test (**d** $p = 0.15$), asterisks indicate significant difference, ***$p < 0.001$. **g** Genetically ablating ASH disrupts learning. Wild-type and ASH-ablated animals are compared using two-sided Welch's *t*-test ($p = 0.0039$), parentheses contain the numbers of assays measured over four independent experiments, circles indicate individual data points, asterisks indicate significant difference, mean ± s.e.m., **$p < 0.01$. Source data are provided as a Source Data file.

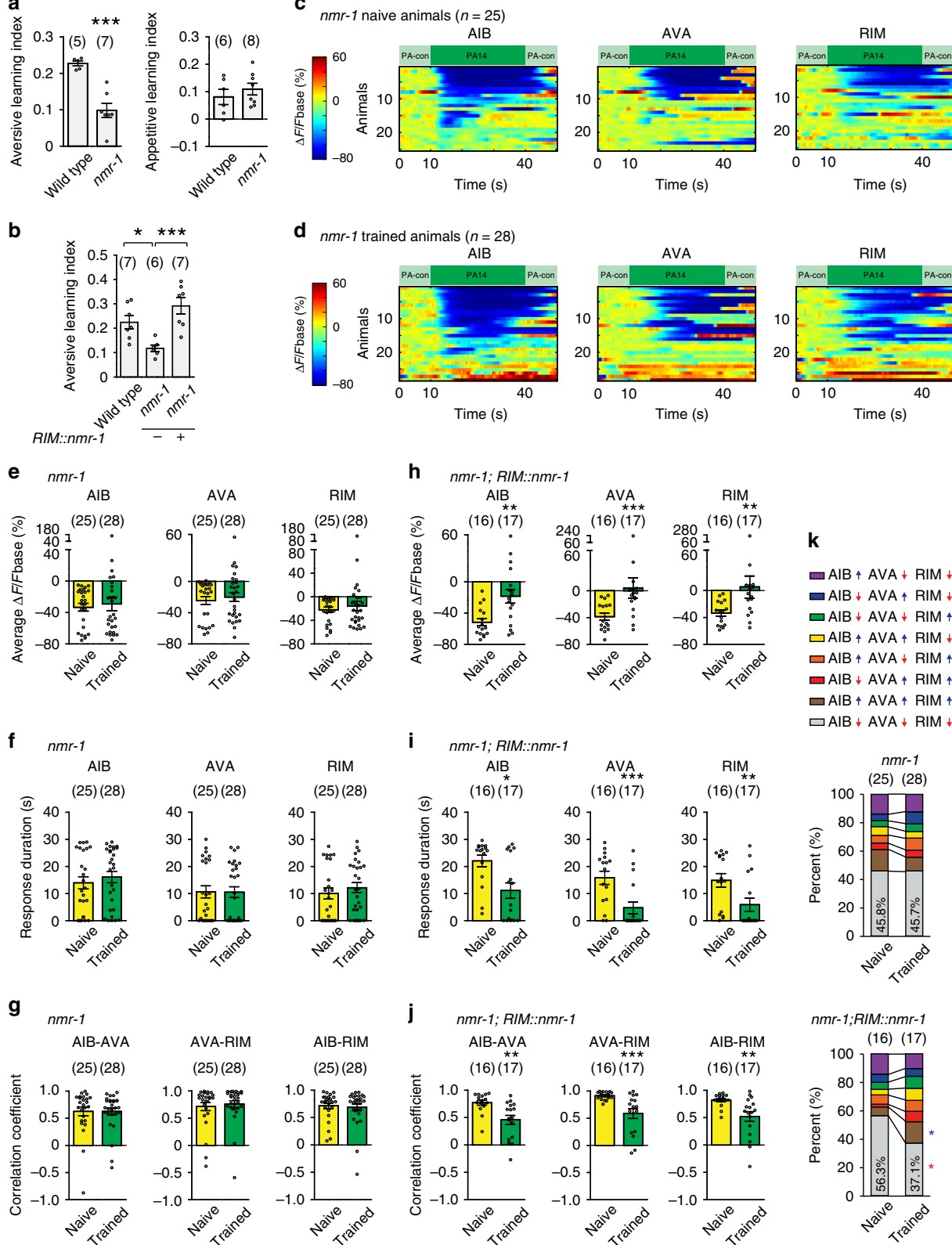

reversals and turns. Training with the pathogenic bacterium PA14 downregulates the gap junction between RIM and AIB, through a RIM-expressed NMDAR that acts through CaMKII to reduce the abundance of a gap junction molecule in a cell-autonomous manner. Training also induces repulsive responses to PA14 in the nociceptive neurons ASH that are presynaptic to

the interneurons without significantly alter the sensory response of the main olfactory sensory neurons AWC to PA14. Thus, RIM-circuit with training-induced weakening of gap junctions receives opposing information of PA14 from AWC and ASH, which generates less correlated PA14-evoked activities in AIB, AVA, and RIM to produce more turns in response to PA14 in trained

**Fig. 5 nmr-1 acts in RIM to regulate learning and training-induced decoupling. a, b** The *nmr-1(ak4)* mutants are defective in aversive learning of PA14, but normal for appetitive learning of OP50 (**a** two-sided Welch's *t*-test), expressing wild-type *nmr-1* cDNA in RIM rescues the defect (**b** one-way ANOVA with Tukey's post hoc test), mean ± s.e.m., parentheses contain the numbers of assays measured over 4 (**a** aversive learning, $p = 0.0003$) or 5 (**a** appetitive learning $p = 0.44$; **b** *p = 0.035, ***p = 0.0009$) independent experiments, circles indicate individual data points. **c, d** GCaMP3 signals of AIB, AVA, RIM evoked by PA14 and control in naive (**c**) and trained (**d**) *nmr-1(ak4)* mutants. GCaMP3 signals of RIM, AIB and AVA are simultaneously recorded in every animal and arranged based on the signals of AIB. $\Delta F = F - Fbase$, Fbase is average GCaMP3 signal in 10s window before PA14 stimulation. Spectrums indicate the ranges of $\Delta F/Fbase$ (%), which are outside of indicated ranges for some frames. **e–k** Average response amplitude (**e**), response duration (**f**), correlation coefficient (**g**), activity pattern (**k**) of PA14-evoked GCaMP3 signals in AIB, AVA, and RIM in *nmr-1(ak4)* mutants are defective in training-induced modulation; expressing wild-type *nmr-1* cDNA in RIM rescues the defects (**h–k**). Average response amplitude and correlation coefficients are for GCaMP3 signals during 30 s PA14 stimulation (**e, g, h, j**), response duration is for responses with $\Delta F/Fbase < -30\%$ (**f, i**). Activity patterns are for the first 20 s PA14 exposure (**k**). In (**k**), upward pointing or downward pointing arrow respectively denotes a positive or negative value for time derivative of GCaMP3 signals. Naive and trained animals are compared using two-sided Welch's *t*-test or two-sided Mann–Whitney *U* test (**e–j**), or two-sided Welch's *t*-test or two-sided Mann–Whitney *U* test with Bonferroni correction (**k**), mean ± s.e.m. (**e–j**) and mean (**k**) are presented, Supplementary Fig. 9 shows mean ± s.e.m. and *p* values for (**k**). Parentheses contain the numbers of animals examined over 6 (**c–g** *nmr-1* in **k**) or 3 (**h–j** rescue in **k**) independent experiments. Circles indicate individual data points (**e–j**). For all, asterisks indicate significant difference, ****p < 0.001, **p < 0.01, *p < 0.05$. The *p* values in the following panels, from left to right, are: **e** 0.68, 0.57, 0.95; **f** 0.55, 0.64, 0.46; **g** 0.61, 0.73, 0.98; **h** 0.0029, 0.0008, 0.0024; **i** 0.01, 0.0008, 0.0029; **j** 0.0025, 0.0008, 0.0067. Source data are provided as a Source Data file.

animals, indicating a reduced preference for PA14 in behavior (Supplementary Fig. 14). Because ASH do not respond to OP50 odorants in either naive or trained animals under our conditions, training specifically modulates PA14-evoked responses in RIM-circuit. While NMDAR-mediated or mGluR-mediated modulation of electrical synapses have been shown in the central nervous system[8–12], our study causally links the modulation of a gap junction with a learning behavior and identifies the cell-autonomous downregulation of the abundance of a gap junction molecule INX-4 by NMDAR and CaMKII as the molecular and cellular underpinning of the modulation.

The NMDARs regulate potentiation or depression of chemical synapses in the mammalian brain. CaMKII acts as a main downstream effector to activate signaling pathways that regulate the AMPA-type glutamate receptors[57,63]. Recent studies on vertebrate brains have shown that stimulating glutamatergic chemical synapses can also potentiate or weaken electrical synapses in the same brain region and in some of the cases blocking the function of NMDAR or CaMKII with pharmacological reagents abolishes the modulatory effects[10–12,64]. These findings reveal the role of NMDAR → CaMKII pathway in regulating the plasticity of gap junctions. *C. elegans* NMR-1 shows a strong identity to the mammalian NMDAR subunit NR1[49,65]. Meanwhile, UNC-43 is the only *C. elegans* homolog of the primary CaMKII in the mammalian brain and shares 69% of sequence identity with CaMKII[58,59]. Using genetics our study demonstrates that NMR-1 and UNC-43 modulate the gap junctions in RIM neurons to regulate learning of food odorants by acting in RIM to reduce the abundance of the gap junction molecule INX-4 in RIM. These findings provide insights into the mechanisms whereby NMDAR and CaMKII modulate gap junctions.

While previous studies have demonstrated dynamic regulation of gap junctions, the underlying molecular mechanisms are not fully understood. In the vertebrate retina, the connexin36-mediated coupling between the AII amacrine cells is modulated by light, which also leads to the phosphorylation of Cx36, a process that can be blocked by pharmacologically blocking the function of CaMKII or NMDAR[66,67], suggesting the phosphorylation of Cx36 by CaMKII as a molecular event underlying the modulation of the electrical synapses. Here, by measuring the level of GFP::INX-4 in RIM cell body, we showed that learning reduced the abundance of INX-4. Because *inx-4* is expressed in multiple neurons that extend their processes into the nerve ring, we cannot identify GFP::INX-4 localized on the RIM process. Nevertheless, our study demonstrates that changing protein abundance is another mechanism through which NMDAR and

CaMKII modulate gap junctions. Regulated protein degradation mediates cellular responses to environmental changes. Particularly, the ubiquitination-mediated protein degradation plays a critical role in synaptic plasticity[68]. In some of these cases, CaMKII acts as a scaffolding molecule to recruit proteasomes to the modulated chemical synapses and activates the proteasomes for protein degradation[69]. Thus, it is plausible that NMDAR and CaMKII reduce the abundance of INX-4 in trained animals through a protein degradation process, such as those mediated by ubiquitination, to decouple PA14-evoked responses in RIM-circuit. These modulatory events alter the sensory-evoked locomotion to display the learned olfactory preference at the behavioral level.

We show that NMR-1 mediates training-dependent down-regulation of INX-4 and that removing *inx-4* rescues the defect of the *nmr-1* mutants in generating training-dependent changes in PA14-induced neuronal responses of RIM-circuit. These findings propose that the glutamatergic neurotransmission-mediated activity of NMR-1 cell-autonomously downregulates the gap junction of RIM. The wiring diagram of the worm nervous system shows that the electrical synapses between RIM and AIB are next to the postsynaptic sites of RIM, through which RIM connect with several glutamatergic neurons, AIB, ASH and ADA, via chemical synapses[15]. The close localizations of the RIM-AIB electrical synapse and these chemical synapses support their functional interactions. Thus, we tested whether ASH was required for the training-dependent downregulation of INX-4. Although training modulates PA14-evoked response of ASH and ASH is required for learning, ablating ASH does not significantly disrupt the downregulation of INX-4 in trained animals. These results suggest that the function of ASH is to generate repulsive response to PA14, which acts together with the weakened gap junctions of RIM to produce decoupled responses to PA14 in the RIM-circuit of the trained animals (Supplementary Fig. 14). The glutamatergic signal that activates NMR-1 to downregulate INX-4 during training is independent of ASH.

Previous studies have shown that RIM, AIB, and AVA exhibit variable sensory-evoked responses that are often correlated with each other. Interestingly, reducing the output signal of RIM chemical synapses increases the robustness of the responses in AIB and AVA that are evoked by a chemical attractant, IAA. These findings suggest that the synaptic release from RIM inhibits the correlation of sensory-evoked responses in RIM-circuit[26]. Here, we find that training- and NMDAR-dependent down-regulation of RIM-gap junctions decouples PA14-evoked sensory responses in RIM, AIB, and AVA, which reveals the function of

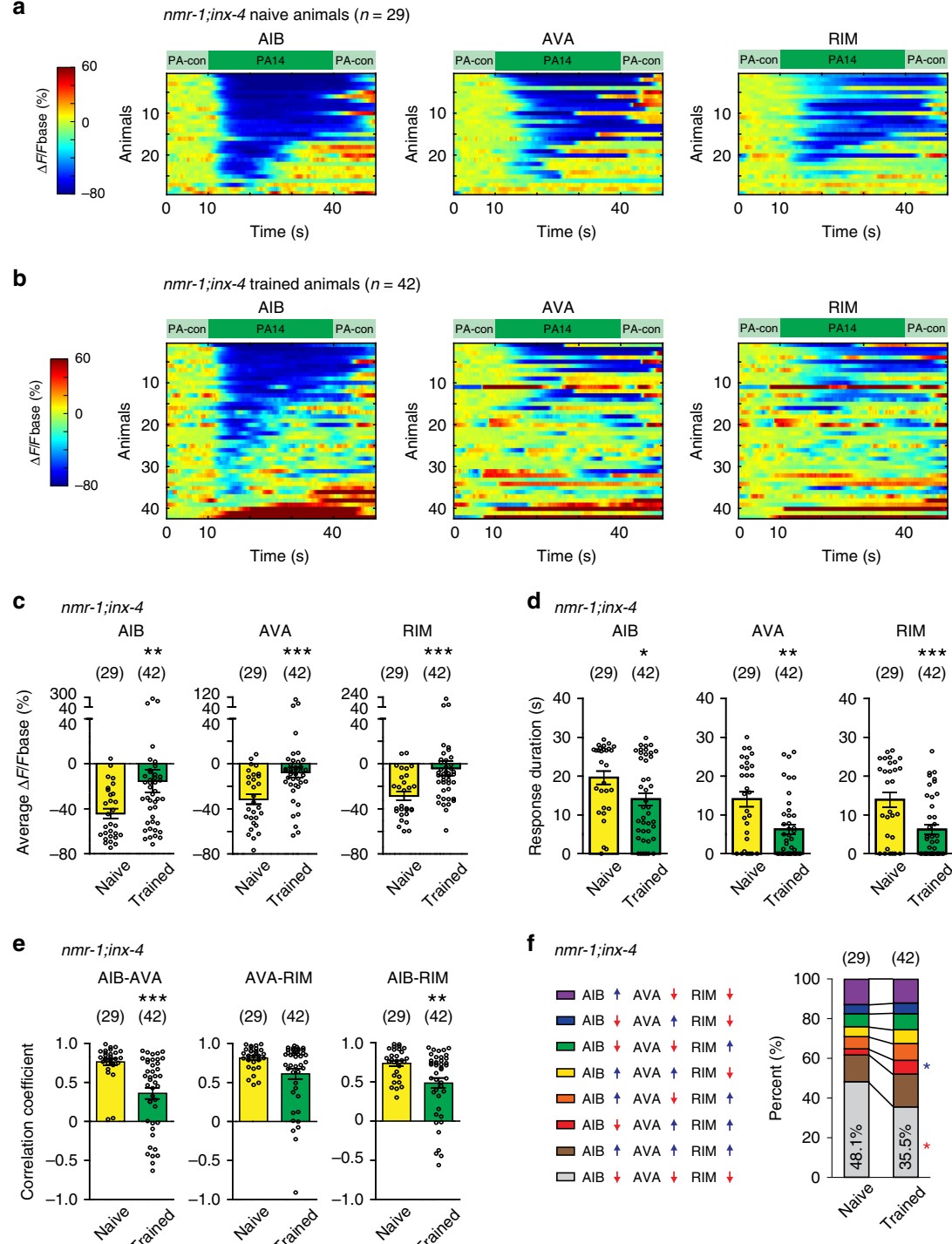

**Fig. 6 inx-4 acts downstream of nmr-1 in RIM to regulate training-induced decoupling. a, b** GCaMP3 signals of AIB, AVA and RIM evoked by switches between PA14 and control in individual naive (**a**) and trained (**b**) nmr-1(ak4);inx-4(ok2373) double mutant animals. GCaMP3 signals of RIM, AIB, and AVA are simultaneously recorded in each animal and arranged based on AIB response. ΔF = F−Fbase and Fbase is average GCaMP3 signal in the 10 s window before PA14 stimulation ("Methods"). Spectrums indicate ranges of ΔF/Fbase (%) and ΔF/Fbase (%) are outside of indicated ranges for some frames. **c–f** Average response amplitude (**c**), response duration (**d**), correlation coefficients (**e**), and activity patterns (**f**) of PA14-evoked GCaMP3 signals in AIB, AVA, and RIM in naive and trained nmr-1;inx-4 double mutant animals. Average response amplitude and correlation coefficients are for GCaMP3 signals during 30 s PA14 stimulation (**c, e**), response duration is for response with ΔF/Fbase < −30% (**d**). Activity patterns are for the first 20 s PA14 exposure (**f**). In **f** upward pointing or downward pointing arrow following each neuron respectively denotes a positive or a negative value for time derivative of GCaMP3 signal of the neuron. Naive and trained animals are compared using two-sided Mann–Whitney U test (**c–e**), or two-sided Welch's t-test or two-sided Mann–Whitney U test with Bonferroni's correction (**f**), mean ± s.e.m. (**c–e**) and mean values (**f**) are presented, Supplementary Fig. 9 shows mean ± s.e.m. and p values for (**f**). For all, parentheses contain the numbers of animals examined over five independent experiments. Circles in **c–e** indicate individual data points, asterisks in **c–f** indicate significant difference, ***p < 0.001, **p < 0.01, *p < 0.05. The p values in the following panels, from left to right, are: **c** 0.0078, 0.0008, 0.0005; **d** 0.0167, 0.0024, 0.0007; **e** <0.0001, 0.070, 0.0046. Source data are provided as a Source Data file.

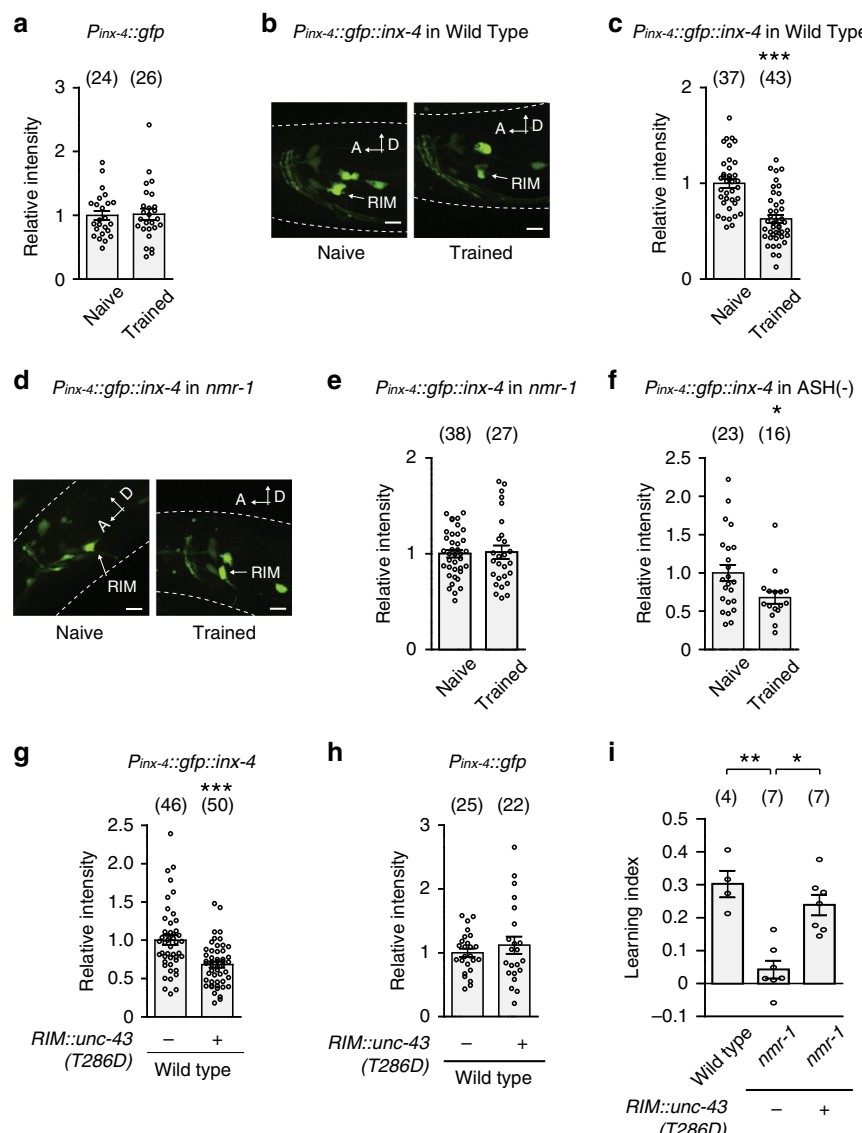

**Fig. 7 *nmr-1* regulates learning by downregulating INX-4 in RIM via UNC-43/CaMKII. a** Training does not alter the expression of $P_{inx-4}::gfp$ transcriptional reporter in RIM, mean ± s.e.m., $p = 0.88$. **b, c** Images of naive and trained transgenic animals expressing $P_{inx-4}::gfp::inx-4$ translational fusion in wild type (**b** similar expression patterns observed in all relevant experiments); and quantitation of fluorescence intensity of $P_{inx-4}::gfp::inx-4$ in RIM (**c** $p < 0.0001$). Arrows denote RIM soma, scale bar: 5 μm. A, anterior; D, dorsal, mean ± s.e.m. **d, e** Images of naive and trained transgenic animals expressing $P_{inx-4}::gfp::inx-4$ translational fusion in *nmr-1(ak4)* mutants (**d** similar expression patterns observed in all relevant experiments); and quantitation of fluorescence intensity of $P_{inx-4}::gfp::inx-4$ in RIM in naive and trained *nmr-1(ak4)* mutant animals (**e** $p = 0.84$). Arrows denote RIM soma, scale bar: 5 μm. A, anterior; D, dorsal, mean ± s.e.m. **f** Quantitation of fluorescence intensity of $P_{inx-4}::gfp::inx-4$ in RIM in naive and trained ASH-ablated animals, mean ± s.e.m., $p = 0.024$. **g, h** Expressing *unc-43(T286D)* in RIM in wild-type animals downregulates the signal of $P_{inx-4}::gfp::inx-4$ translational fusion in RIM (**g** $p < 0.0001$) without altering the expression of $P_{inx-4}::gfp$ transcriptional fusion in RIM (**h** $p = 0.43$), mean ± s.e.m. For (**a, c, e–h**), GFP intensity is normalized by average intensity of naive (**a, c, e, f**) or non-transgenic (**g, h**) control animals; naive animals are compared with trained animals or transgenic animals (+) are compared with non-transgenic controls (−) using two-sided Welch's *t*-test or two-sided Mann–Whitney *U* test. Parentheses contain the numbers of animals examined over 3 (**a, c, e**) or 2 (**f–h**) independent experiments, circles indicate individual data points. **i** Expressing *unc-43(T286D)* in RIM suppresses the learning defect in *nmr-1(ak4)* mutant animals. Wild-type, non-transgenic (−), and transgenic (+) mutant animals are compared using Kruskal–Wallis test with Dunn's multiple comparison test (**$p = 0.0063$, *$p = 0.021$). Parentheses contain the numbers of assays measured over four independent experiments, circles indicate individual data points. For all, asterisks indicate statistical difference, ***$p < 0.001$, **$p < 0.01$, *$p < 0.05$. Source data are provided as a Source Data file.

the RIM-gap junctions in facilitating correlated sensory responses in RIM-circuit. At the behavioral level, previous results have shown that disrupting synaptic outputs of RIM abolishes olfactory learning of PA14[34]. Here, we find that learning of PA14 requires the downregulation of RIM-gap junctions. These findings together reveal distinct functions for the chemical synapses versus the NMDAR-dependent modulation of electrical synapses in a pair of central neurons in regulating the plasticity of neural circuit and behavior.

## Methods

**Strains**. The adult *Caenorhabditis elegans* hermaphrodites were used in this study. The *C. elegans* strains were cultivated and maintained using standard conditions[37]. The strains used include: N2, CX14996 *kyEx4965[P_{inx-1}::GCaMP3, P_{tdc-1}::GCaMP3,*

$P_{rig-3}$::GCaMP3, $P_{unc-122}$::dsRed], RB1834 inx-4(ok2373)V, VM487 nmr-1(ak4)II, ZC2787 nmr-1(ak4)II; yxEx1440[$P_{gcy-13}$::nmr-1cDNA, $P_{unc-122}$::gfp], JN1713 peIs1713[$P_{sra-6}$::mCasp-1; $P_{unc-122}$::mCherry], ZC2671 yxEx1371[$P_{odr-1}$::GCaMP6s, $P_{str-2}$::mCherry, $P_{unc-122}$::gfp], ZC2905 yxEx1508[$P_{sra-6}$::GCaMP6s, $P_{unc-122}$::dsRed], ZC2788 nmr-1(ak4)II; inx-4(ok2373)V, ZC2947 yxEx1531[$P_{gcy-13}$::Cx36::mCherry, $P_{nmr-1s}$::LoxPSTOPLoxP::Cx36::mCherry, $P_{flp-18}$::nCre, $P_{inx-1}$::Cx36::mCherry, $P_{unc-122}$::gfp], ZC2948 yxEx1532[$P_{gcy-13}$::Cx36::mCherry, $P_{inx-1}$::Cx36::mCherry, $P_{unc-122}$::gfp], ZC2950 yxEx1534[$P_{gcy-13}$::Cx36::mCherry, $P_{nmr-1s}$::LoxPSTOPLoxP::Cx36::mCherry, $P_{flp-18}$::nCre, $P_{unc-122}$::gfp], ZC2952 nmr-1(ak4)II; kyEx4965, ZC2953 nmr-1(ak4)II; inx-4(ok2373)V; kyEx4965, ZC2954 yxEx1531; kyEx4965, ZC2955 nmr-1(ak4)II; yxEx1440; kyEx4965, ZC2956 yxEx1536[$P_{inx-4}$::gfp::inx-4 cDNA(isoform c), $P_{gcy-13}$::mCherry, $P_{unc-122}$::gfp], ZC2957 nmr-1(ak4)II; yxEx1536, ZC2958 yxEx1537[$P_{inx-4}$::gfp, $P_{gcy-13}$::mCherry, $P_{unc-122}$::gfp], ZC2959 nmr-1(ak4)II; yxEx1538[$P_{gcy-13}$::unc-43(T286D)::mCherry, $P_{unc-122}$::dsRed], ZC2960 yxEx1536; yxEx1538, ZC3326 inx-4(ok2373)V; yxEx1536, ZC3327 nmr-1(ak4)II; inx-4(ok2373)V; kyEx4965; yxEx1535[$P_{gcy-13}$::inx-4cDNA, $P_{unc-122}$::gfp], ZC3328 peIs1713; yxEx1536.

**Transgenes and transgenic animals**. To make a destination vector for nmr-1 cDNA, the Gateway recombination cassette (Invitrogen) was inserted into pPD49.26 (a gift from A. Fire) and the nmr-1 cDNA was cloned downstream of the cassette. The destination vector containing the inx-4 cDNA or the mammalian Connexin36 cDNA fused with the sequence of mCherry [a gift from Schafer[43]] was similarly generated. The 2.2 kb region upstream of gcy-13[42] or the 0.9 kb region upstream of inx-1[19] was inserted into pCR8 Gateway entry clone (Invitrogen) to drive neuron-specific expression in RIM or AIB, respectively. AVA specific expression of Cx36 was driven by co-expression of $P_{flp-18}$::nCre[31] and $P_{nmr-1sp}$::LoxPSTOPLoxP::Cx36::mCherry, the latter of which was generated by inserting $P_{nmr-1sp}$::LoxPSTOPLoxP into pCR8 and performing LR recombination with the Cx36::mCherry destination vector. The 3.5 kb region upstream of inx-4 was inserted into pCR8 to generate an entry vector, which was recombined with destination vector containing gfp or inx-4 cDNA to generate $P_{inx-4}$::gfp or Pinx-4::inx-4 cDNA, respectively. Pinx-4::gfp::inx-4 cDNA was generated by inserting the gfp sequence from pPD95.95 (a gift from A. Fire) into Pinx-4::inx-4 cDNA. $P_{gcy-13}$::unc-43(T286D)::mCherry was generated by Gibson assembly (NEB) and Q5 site-directed mutagenesis (NEB). The transgenic animals were generated with microinjection[70] with appropriate transgenes at 50 ng/μL using $P_{unc-122}$::gfp or $P_{unc-122}$::dsRed as co-injection marker at 30 ng/μL. The primers used in this study are listed in the Supplementary Methods.

**Aversive olfactory training and learning assay**. The aversive training with pathogenic bacteria and the learning assay (Supplementary Fig. 1) were performed using established protocols[23]. Young adult animals cultivated under the standard condition were transferred to a nematode growth medium plate (NGM, NaCl 3 g/L, peptones 2.5 g/L, 1 mM CaCl$_2$, 1 mM MgSO$_4$, 25 mM KPO$_4$ pH6.0, 1.6% Agar, Cholesterol 5 mg/L) covered by a fresh lawn of E. coli OP50 (naive control) or P. aeruginosa PA14 (aversive training) generated by incubation at 26 °C for two days and trained at 22 °C for 4–6 h. The olfactory preference of naive animals and trained animals were tested in a droplet assay or a chemotaxis assay on a NGM plate[36,44]. Cultures of OP50 and PA14 in the NGM medium generated by overnight incubation at 26 °C were used to produce olfactory stimuli. In the droplet assay, bacterial cultures were used directly or diluted with NGM medium[23,36]. The swimming behavior of the tested animals were recorded and large body bends were measured by computer softwares[23,36]. The choice index and learning index are defined as: Choice index for PA14 = (turning rate to PA-control − turning rate to PA14)/(turning rate to PA-control + turning rate to PA14); Choice index for OP50 = (turning rate to OP-control − turning rate to OP50)/(turning rate to OP-control + turning rate to OP50); Aversive learning index = choice index of naive animals − choice index of trained animals; Appetitive learning index = choice index of trained animals − choice index of naive animals. In the plate assay, a drop of 10 μL PA14 culture supernatant was placed in the center of an empty NGM plate. After briefly crawling on another empty NGM plate to remove the bacteria on the body, an individual worm was placed 1.5 cm away from the center of the bacterial supernatant. The chemotactic movements towards PA14 were recorded at 7 Hz right afterwards (note: some time may pass between the transfer of the worm and the beginning of the recording; FLIR Integrated Imaging Solution). The recordings were analyzed with the WormLab System (MBF Bioscience) and previously published methods. Navigation index is the ratio between the radial speed and the traveling speed, which indicates the efficiency of moving towards PA14[44].

**Calcium imaging**. Calcium imaging was performed with the aid of a microfluidic device[27] using previously established protocols[23,44]. Specifically, to prepare OP50 or PA14 conditioned medium, NGM culture of OP50 or PA14 was generated by overnight incubation at 26 °C and then span down using a tabletop centrifuge for 20 min at 3320 × g. The supernatant was used as the conditioned medium. The fluorescence time-lapse imaging was collected at 5 frames per second on a Nikon Eclipse Ti-E inverted confocal microscope with an ANODR iXon Ultra EMCCD camera using a ×40 oil immersion objective. The movies were analyzed with ImageJ v1.50b (NIH). Briefly, a region of interest (ROI) was selected manually based on the anatomical position of the neuron and tracked throughout the movie using ImageJ. The fluorescence intensity (F) of each ROI in each frame was generated by

subtracting the average intensity of the ROI with the average signal intensity of an background area of the similar size and shape. The change of the fluorescence intensity (ΔF) was calculated by subtracting F with the average intensity of the 10 s window before the stimulation of PA14 or OP50 (Fbase). ΔF/Fbase was calculated for each frame and displayed using basic functions in Matlab. Imaging of 50 s is shown for each movie in the Figures. Three different thresholds, ΔF/Fbase = −10% or −30% or −50%, were used to calculate response duration for RIM, AIB and AVA, which is defined as the duration when ΔF/Fbase < −10%, −30% or −50% (as shown in Supplementary Fig. 2). Correlation coefficients were generated in Excel v16.0 using ΔF/Fbase of each frame. To analyze the activity pattern of AIB, AVA and RIM neurons, we binned the GCaMP3 signals for each second and calculated the instantaneous time derivative for every second. An upward pointing arrow or a downward pointing arrow following each neuron in the relevant text and figures denotes a positive or a negative value for the time derivative of the GCaMP3 signal of the neuron, which respectively indicates an increase or a decrease in neuronal activity. The percentage of time when each activity pattern is displayed by the circuit during the 20 s window after the onset of stimulation is used to analyze the activity patterns of the interneuronal circuit, because in naive wild type the changes in calcium signals become stable after about 20s-exposure to PA14. The analysis on PA-control was done using the 10 s window before switching from PA-control to PA14. The statistical analyses were performed using GraphPad Prism v8.4.2.

**Confocal microscopy**. To measure fluorescent intensity of the $P_{inx-4}$::gfp::inx-4 translational fusion or the $P_{inx-4}$::gfp transcriptional fusion in RIM, Z stack images were collected on a Nikon Eclipse Ti-E inverted confocal microscope from each worm to generate the maximum intensity projection, which was analyzed using NIH ImageJ v1.50b. The intensity of the GFP or GFP::INX-4 signal in RIM soma was measured by subtracting the average intensity of a ROI, determined based on the signal of a co-transformed mCherry transgene selectively expressed in RIM, by the average intensity of the background signal of the same size and shape. Multiple worms were analyzed on different days for each condition.

**Quinine sensitivity assay**. The sensitivity to quinine was tested based on previously published procedures[56]. Specifically, individual young adult worms were tested on an empty NGM plate, 10–20 min after being moved onto the plate. Quinine HCL (Sigma-Aldrich) was dissolved in M13 buffer and a small drop of 1 mM quinine solution was placed in front of a worm using a glass needle and a mouse pipette. If a worm reversed from the quinine solution within 4 s after encountering the solution, it was considered as a positive response. Each worm was tested for three times with 5 min intervals. The number of positive responses out of three trials was recorded as the avoidance index for the worm. Multiple worms were tested for each genotype.

**Statistical analysis**. The statistical tests were performed using GraphPad Prism v8.4.2. and imaging analysis was done using ImageJ v1.50b. The statistical tests, value of n and what each n value represents, and other related measures are shown in the legend of each relevant figure. In all, asterisks denote significant difference, ***p < 0.001, **p < 0.01, *p < 0.05.

**Reporting summary**. Further information on research design is available in the Nature Research Reporting Summary linked to this article.

## Data availability
All data generated or analyzed during this study are included in the article and its supplementary information files. Source data are provided with this paper.

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

## Acknowledgements

We thank *Caenorhabditis Genetics Center* funded by NIH Office of Research Infrastructure Programs P40 OD010440 and the *C. elegans* Gene Knockout Project at the Oklahoma Medical Research Foundation, which was part of the International *C. elegans* Gene Knockout Consortium, for providing strains. We thank Drs Bargmann C, Maricq V for strains, Achala Chittor for helping with behavioral assays, and Zhang laboratory members for critically discussing the manuscript. M.C. was supported by Basic Science Research Program through the National Research Foundation of Korea (NRF) funded by the Ministry of Education (2014R1A6A3A03060041) for 2014 -2015. The work in the Zhang lab is supported by National Institutes of Health (DC009852).

## Author contributions

M.C. and Y.Z. conceived the project, interpreted the results and wrote the manuscript. M.C., H.L. T.W., and W.Y. designed and performed experiments and analyzed data.

## Competing interests

The authors declare no competing interests.
