## [Peer Review File · Nature Communications]

Reviewers' Comments:

Reviewer #1:

Remarks to the Author:

Choi et al. present an extensive neurogenetic analysis of a form of gap junction-mediated plasticity underlying aversive olfactory conditioning in *C. elegans*. The authors describe a gap junction circuit consisting of 3 interneurons (AIB, AVA, and RIM) whose correlated suppression mediates decreased turning (related to attractive navigation behavior) to a pathogenic bacteria PA14 in naïve animals. Following training, this correlated suppression is disrupted, causing an increase in turning responses to PA14 (related to avoidance). The authors also show that training with PA14 causes the nociceptive neuron ASH to respond to PA14 and act on this gap junction circuit to promote reversals. Intriguingly, ASH does not respond to PA14 in naïve animals. The authors further discover that this plasticity is mediated by weakened gap junction connections between two of these neurons (RIM-AIB), and that this decreased strength is due to reduced abundance of a specific gap junction protein *inx-4*. Further, the authors show that NMDA receptors acting through CaMKII control plasticity induced changes in *inx-4* levels. These findings are all novel and/or represent an important *in vivo* validation of previously described plasticity mechanisms. The observation that PA14 training recruits responses from ASH will further stimulate additional studies in this area.

In sum, this work is technically impressive and an important advance that uses many of the advantages of the *C. elegans* system to validate and extend our understanding of a conserved form of neural circuit plasticity.

There are only a few concerns that should be addressed:

-It is concerning to use omega bends as a proxy for odorant preference when it is possible to measure preference behavior directly with a simple chemotaxis assay. There are many potential confounding factors that could increase stimulus evoked bending without truly altering stimulus preference. This makes claims like "We found that, strikingly, the *nmr-1* mutants that expressed UNC-43(T286D) in RIM learned to avoid PA14 similarly as wild-type animals (Fig. 7h)" misleading as only bending/turning rates were quantified, not true behavioral avoidance/choice. This is a particularly important consideration here, as the authors manipulate *nmr-1* (involved in reversals for local search/foraging behavior) and *innexins* (which can cause non-specific behavioral impairments), and CaMKII (an uncoordinated mutant critical for proper neurodevelopment and many diverse neuronal functions).

The authors should validate the critical mutant and rescue behavioral findings with one of the true PA14 conditioning approaches they have developed in previous work. Specifically, it would be ideal to see the *nmr-1* mutant and rescue lines, *nmr-1;inx-4* double, and *unc-43(T286D)* findings validated in a binary choice chemotaxis behavioral assay.

-The manuscript would benefit from extensive editing by a native English speaker. E.g. the first sentence is grammatically incorrect as are many others. For example "We previously show that *C. elegans* learns to avoid" should read "We have previously shown" or "there are totally eight different patterns of the activity" should read "In total there are eight possible activity patterns"

Reviewer #2:

Remarks to the Author:

In this work by Choi et al, the authors investigate the role gap junctions play in neural plasticity in learning, using *C. elegans* as a model system. The authors leveraged the simplicity and capability to manipulate defined circuits in the worm to show how electrical pairings between specific pairs of neurons could be modified by experience. Specifically, the authors showed how the expression of *INX-4* in the neuron RIM could be regulated through activation of NMDAR and subsequent

signaling by CaMKII, to down-regulate the expression of INX-4, and thereby decrease the coupling between neurons in an olfactory circuit. This regulation appears to occur as a result of the aversive experience of consuming PA-14, a pathogen which is initially attractive to the worm, but subsequently becomes repulsive after the worm digests it (training), and learns to avoid it. The resulting change in the olfactory circuit that regulates attraction leads to the worms decreasing chemotaxis behavior that would drive them toward the odor source. The authors were also able to use an orthogonal approach of expressing connexins in specific pairs of neurons, and show how increased coupling improved responses to attractive odors, regardless of prior training.

I enjoyed reading this work. It took advantage of several techniques available to the authors to manipulate specific neurons, and show how explicit manipulations in a defined circuit could influence learning behavior. However, there were a few key points that confused/concerned me, outlined below:

Calcium Heatmaps: Are the individual trials ordered according to the response to odor? The text says ordered according to animal, but there tends to be a trend of decreasing responses in the animals in subsequent observations. Is this experimental, biological, or are the data ordered according to their response? If the data are not pre-ordered, why does this progression exist? Are the animals becoming less attractive to PA14 over the course of the experiment?

Figure 2 b-e (and similar figures): The term "response time" usually refers to the length of time it takes for a system to change states, or "respond" to a stimulus. However, it appears the metric being measured here is the duration of the response, rather than the response time. Am I correct in thinking this? If so, the wording should be changed.

Time derivative: It is calculated in one second bins for 20 seconds after the odor switch, and then the percentage of time spent in each state is calculated for each of the 20 one-second intervals. Why 20 seconds? This analysis also assumes the derivatives are always positive or negative, but there appear to be long periods where the derivative is close to zero. Is any threshold used for a minimum time derivative? Is most of the variance observed due to states with low time-derivative amplitudes? Not applying a threshold to these categorizations will produce states that are simply the result of noise.

There seems like better ways of doing this:

- 1) Plot the cross-correlation of the neurons in time with a sliding window, and see if correlation increases or decreases when odor is added.
- 2) Actual response time: The time it takes for the neuron's time-derivative to go below a threshold that would indicate it "responded" to the stimulus. This threshold could involve the amplitude of the time-derivative, the length of the response before the time-derivative becomes negative, or both.

The results in figure 3h-k are really nice.

The results in figure 6 are nice.

Figure 7: Why is the GFP::INX-4 protein regulated differently than the GFP protein? The authors claim it's likely not due to differences in transcription, so I'm assuming it's either translational or degradation regulation? They quantify fluorescence in the soma, but wouldn't the native signal be in the process? Are they quantifying differences in ER-localized innexin, or is their construct mislocalized? Was any difference in expression observed in the processes where the innexin should be localized?

It seems like the ASH/AWC story is parallel to the AIB-RIM-AVA story, but there appears to be a lack of connection between the two. How does the increase of PA14-induced activity in ASH lead to

the activation of NMR-1 in RIM? Is it direct activation of RIM by ASH? Presumably ASH is still getting activated in the Cx36 background, but the AWC signal is dominating in this context?

I think the ASH/AWC story is interesting, but it doesn't seem to directly connect to the AIB-RIM-AVA story which is investigated more thoroughly. Also, with regard to AWC, it appears that while the hyperpolarized state of AWC in the trained background is unaltered, it seems like the rate of decay is slower in the trained context.

Overall, I think this is a good study, but I think some re-analysis would help communicate the results better, and I think the ASH/AWC story should be dropped, or how it relates to the AIB-RIM-AVA story should be fleshed out some more.

Reviewer #3:

Remarks to the Author:

In this elegant study, the authors dissect how olfactory learning is controlled by the modulation of gap junction expression at single neuron resolution. Using neuron-specific calcium imaging, behavioral analysis and gap junction expression measurements they show that the experience of learning dampens gap junction protein expression by a conserved regulatory pathway. As such, their study reveals how regulation of gap junction protein levels can lead to experience-dependent changes in behaviour. The research findings are novel and will be of interest to the wider field. The manuscript is very well written and the extensively controlled data are meticulously presented and fully support their conclusions.

However, the following areas need to be made more clear:

Line 181-183 - the rationale leading to this hypothesis below is not strong. Please revise. "We hypothesized that the training-induced changes in PA14-evoked activities of AIB, AVA and RIM, such as the decoupling of the activities, resulted from weakening of the gap junctions."

Line 222-223 - The sentence containing this ("do not change their response to PA14 odorants") is not clear. This sentence reads that the AWCs do not respond to PA14 odorants at all but the calcium imaging clearly shows that PA14 odor suppresses AWC activity. Therefore, the AWCs do change their response to PA14 but not to training.

Line 224 - There seems to be some residual learning upon ASH ablation. Perhaps the term "completely abolished" needs to be revised in the following sentence? "Genetically ablating ASH completely abolished the aversive learning of PA14 (Fig 4g).

Line 232 - the rationale behind examining the role of NMR-1 is weak. The authors must introduce this section with more clarity.

I also identified a few typographical that need to be amended:

Line 58 - remove "the" after "detect"

Line 73 - change "show" to "showed" after previously

Line 79 - remove "the" before "training-dependent" and before "RIM gap"

Line 154 - change "totally" to "in total" before "eight"

Line 297 - change "somata" to "soma"

Line 357 - change "systems" to "system"

The greek omega sign should be defined the first time it is mentioned.

Point-by-point response to the reviews

We appreciate the reviewers for giving us instructive comments and suggestions. We have performed new experiments and new data analyses as the reviewers suggested. The newly generated results support our original findings and further strengthen the conclusions of our study. Please refer to the following point-by-point response to the reviews for the details of our revision.

Reviewers' comments:

Reviewer #1 (Remarks to the Author):

Choi et al. present an extensive neurogenetic analysis of a form of gap junction-mediated plasticity underlying aversive olfactory conditioning in *C. elegans*. The authors describe a gap junction circuit consisting of 3 interneurons (AIB, AVA, and RIM) whose correlated suppression mediates decreased turning (related to attractive navigation behavior) to a pathogenic bacteria PA14 in naïve animals. Following training, this correlated suppression is disrupted, causing an increase in turning responses to PA14 (related to avoidance). The authors also show that training with PA14 causes the nociceptive neuron ASH to respond to PA14 and act on this gap junction circuit to promote reversals. Intriguingly, ASH does not respond to PA14 in naïve animals. The authors further discover that this plasticity is mediated by weakened gap junction connections between two of these neurons (RIM-AIB), and that this decreased strength is due to reduced abundance of a specific gap junction protein *inx-4*. Further, the authors show that NMDA receptors acting through CaMKII control plasticity induced changes in *inx-4* levels. These findings are all novel and/or represent an important *in vivo* validation of previously described plasticity mechanisms. The observation that PA14 training recruits responses from ASH will further stimulate additional studies in this area.

In sum, this work is technically impressive and an important advance that uses many of the advantages of the *C. elegans* system to validate and extend our understanding of a conserved form of neural circuit plasticity.

We appreciate the reviewer's comments on our manuscript.

There are only a few concerns that should be addressed:

-It is concerning to use omega bends as a proxy for odorant preference when it is possible to measure preference behavior directly with a simple chemotaxis assay. There are many potential confounding factors that could increase stimulus evoked bending without truly altering stimulus preference. This makes claims like "We found that, strikingly, the *nmr-1* mutants that expressed UNC-43(T286D) in RIM learned to avoid PA14 similarly as wild-type animals (Fig. 7h)" misleading as only bending/turning rates were quantified, not true behavioral avoidance/choice. This is a particularly important consideration here, as the authors manipulate *nmr-1* (involved in reversals for local search/foraging behavior) and *innexins* (which can cause non-specific behavioral impairments), and CaMKII (an uncoordinated mutant critical for proper neurodevelopment and many diverse neuronal functions).

The authors should validate the critical mutant and rescue behavioral findings with one of the true PA14 conditioning approaches they have developed in previous work. Specifically, it would be ideal to see the *nmr-1* mutant and rescue lines, *nmr-1;inx-4* double, and *unc-43(T286D)* findings validated in a binary choice chemotaxis behavioral assay.

We understand the reviewer's concern. Previously, we showed that by quantifying the bending/turning rate, the droplet assay measured the behavioral response to several commonly used odorants, such as benzaldehyde and isoamyl alcohol, similarly as the standard chemotaxis assay performed using plates (Luo et al., *J Neurophysiology* 2008). To establish the droplet assay as a way to quantify olfactory learning, we also

showed that the olfactory preference displayed in the droplet assay and in the two-choice assay, as well as the molecular and cellular mechanisms underlying olfactory learning identified by these two types of assays were consistent (Ha et al., Neuron 2010). However, we agree with the reviewer that in the droplet assay the worms do not actually “approach” the odorant sources and that it is helpful to strengthen our findings with an assay that measures the movement of the worms to the odorant source.

Thus, to address the concern, we used a previously established assay in which a worm crawled towards a drop of fresh culture of PA14 on a plate. The locomotion of the worm was recorded and quantified (Liu et al., Neuron 2018). We previously showed that training adult worms with PA14 for 4-6 hours (the same as in this study) reduced the preference of the worms for PA14, demonstrated by the slower and less efficient crawling towards PA14 in trained worms in this plate assay (Supplementary Fig. 10a and Liu et al., Neuron 2018).

In our original manuscript, using the droplet assay we showed that the *nmr-1* mutant animals and the transgenic animals ectopically expressing connexin Cx36 in AIB, RIM and AVA were both defective in aversive learning of PA14. Now, using the plate assay, we show that both of these mutant animals are similarly defective in learning of PA14 (Supplementary Fig. 10b, 10c, and 10d). Furthermore, using the droplet assay we showed that expressing a wild-type *nmr-1* cDNA in RIM or expressing UNC-43(T286D) in RIM rescued the learning defect of *nmr-1*. Similarly, expressing these transgenes rescues the learning defect of the *nmr-1* mutant animals in the plate assay (Supplementary Fig. 10e, 10f, 10g and 10h). Together, these results validated the behavioral findings in these animals that we reported in the original manuscript.

We did not test learning in the *nmr-1;inx-4* double mutants in our original manuscript, because *inx-4* is widely expressed and the *inx-4* single mutant animals are uncoordinated in locomotion. To address the function of the INX-4 gap junction, we showed that (1) mutating *inx-4* in the *nmr-1* mutant animals suppressed the defects of the *nmr-1* mutants in generating learning-dependent modulation of the activities of RIM, AIB, and AVA neurons, (2) the abundance of INX-4 in RIM was downregulated by learning, (3) the *nmr-1* mutation abolished the downregulation of INX-4 in RIM by learning, (4) ectopically expressing the gap junction molecule Cx36 in RIM and AIB disrupted learning. To further confirm the role of *inx-4* in RIM, in the revised manuscript we include the calcium imaging experiments in transgenic animals that express the wild-type *inx-4* cDNA selectively in RIM in the *nmr-1;inx-4* double mutant animals. We show that RIM-specific restoration of *inx-4* in the double mutants reverses the suppressing effect of mutating *inx-4* on the *nmr-1* mutant animals (Supplementary Fig. 12a – 12f).

-The manuscript would benefit from extensive editing by a native English speaker. E.g. the first sentence is grammatically incorrect as are many others. For example “We previously show that *C. elegans* learns to avoid” should read “We have previously shown” or “there are totally eight different patterns of the activity” should read “In total there are eight possible activity patterns”

We apologize for the grammatical errors. We have now carefully edited the manuscript.

Reviewer #2 (Remarks to the Author):

In this work by Choi et al, the authors investigate the role gap junctions play in neural plasticity in learning, using *C. elegans* as a model system. The authors leveraged the simplicity and capability to manipulate defined circuits in the worm to show how electrical

pairings between specific pairs of neurons could be modified by experience. Specifically, the authors showed how the expression of INX-4 in the neuron RIM could be regulated through activation of NMDAR and subsequent signaling by CaMKII, to down-regulate the expression of INX-4, and thereby decrease the coupling between neurons in an olfactory circuit. This regulation appears to occur as a result of the aversive experience of consuming PA-14, a pathogen which is initially attractive to the worm, but subsequently becomes repulsive after the worm digests it (training), and learns to avoid it. The resulting change in the olfactory circuit that regulates attraction leads to the worms decreasing chemotaxis behavior that would drive them toward the odor source. The authors were also able to use an orthogonal approach of expressing connexins in specific pairs of neurons, and show how increased coupling improved responses to attractive odors, regardless of prior training.

I enjoyed reading this work. It took advantage of several techniques available to the authors to manipulate specific neurons, and show how explicit manipulations in a defined circuit could influence learning behavior.

We appreciate the reviewer's comments.

However, there were a few key points that confused/concerned me, outlined below:

Calcium Heatmaps: Are the individual trials ordered according to the response to odor? The text says ordered according to animal, but there tends to be a trend of decreasing responses in the animals in subsequent observations. Is this experimental, biological, or are the data ordered according to their response? If the data are not pre-ordered, why does this progression exist? Are the animals becoming less attractive to PA14 over the course of the experiment?

We apologize for the confusion. As the reviewer pointed out, the calcium imaging data are ordered based on the response of AIB neurons. We have clarified the relevant text and legends.

Figure 2 b-e (and similar figures): The term "response time" usually refers to the length of time it takes for a system to change states, or "respond" to a stimulus. However, it appears the metric being measured here is the duration of the response, rather than the response time. Am I correct in thinking this? If so, the wording should be changed.

The reviewer is correct and we appreciate the comment. We have changed the term "response time" to "response duration" throughout the manuscript and the figures.

Time derivative: It is calculated in one second bins for 20 seconds after the odor switch, and then the percentage of time spent in each state is calculated for each of the 20 one-second intervals. Why 20 seconds? This analysis also assumes the derivatives are always positive or negative, but there appear to be long periods where the derivative is close to zero. Is any threshold used for a minimum time derivative? Is most of the variance observed due to states with low time-derivative amplitudes? Not applying a threshold to these categorizations will produce states that are simply the result of noise.

We appreciate the reviewer's comments. We analyzed the circuit state using the signals for the 20-second exposure after the switch of the stimulus, because on average the changes in the calcium transients of the neurons in the naive wild-type animals became stable after about 20s exposure to PA14 or OP50. We have added this clarification to the Methods section.

We agree with the reviewer that the 20-second window is arbitrary. To address this concern, we did the same analysis on the calcium signals in 0 – 15 second, 0 - 20 second, 0 – 30 second and 5 – 15 second time windows after odorant switch and

reached the same conclusion (Supplementary Fig. 9b, 9c, 9d, 9e). Therefore, the conclusion generated from the state analysis does not depend on the specific time window for which the calcium signals are analyzed.

We did not use a threshold for the time derivatives, because the circuit states change significantly after training (please see the heat maps in Fig. 1e and 1f) and we found that applying a time derivative threshold to the data will differently impact the data of naive animals versus trained animals. In addition, if the states mainly represent the noise in the signals, the noise would randomly contribute to the eight different states. This is not what we saw. We found that training significantly reduced the state of “AIB↓AVA↓RIM↓” and significantly increased the states of “AIB↑AVA↑RIM↓” and “AIB↓AVA↑RIM↓” and that the results were robust when we analyzed the signals in the 0 – 15 second, 0 - 20 second, 0 – 30 second and 5 – 15 second time windows after odorant switch. We have included these additional analyses in Supplementary Fig. 9b, 9c, 9d, and 9e.

Also importantly, we performed the state analysis using the same method on the GCaMP3 signals during PA-control (Fig. 2g and Supplementary Fig. 9a) and on OP50-evoked GCaMP3 signals (Fig. 2g and Supplementary Fig. 9f) and found no difference between naive and trained animals. In addition, the results of the state analysis are consistent with the results of behavioral analysis (Fig. 2h). The activities of RIM, AIB and AVA are correlated with reorienting movements, such as reversals and turns. Activating any of the three pairs of the neurons increases reversals (Gordus et al., Cell 2015). Therefore, the training-dependent decrease in the state in which RIM, AIB and AVA are all suppressed by exposure to PA14 is consistent with the increased turning rate in response to PA14 in the trained animals (Fig. 2h). Thus, although we cannot completely exclude noise from these analyses, these results together show that the state analysis characterizes the calcium responses of the neurons.

There seems like better ways of doing this:

1) Plot the cross-correlation of the neurons in time with a sliding window, and see if correlation increases or decreases when odor is added.

We appreciate the reviewer’s suggestion. We have now plotted the cross-correlation of the neurons with a sliding window. The result shows that the correlation increases when PA14 stimulus is added in naive animals. Also importantly, it shows that the increase of correlation is significantly decreased in trained animals, further supporting our finding that the aversive training decreases the correlated activities of the neurons. The results are now shown in Supplementary Fig. 7a and 7b.

2) Actual response time: The time it takes for the neuron’s time-derivative to go below a threshold that would indicate it “responded” to the stimulus. This threshold could involve the amplitude of the time-derivative, the length of the response before the time-derivative becomes negative, or both.

We are interested in the duration when the neuronal activities are suppressed by PA14 exposure. We agree with the reviewer that the time-derivative below certain threshold would be useful to analyze the calcium signals in the naive animals and some of the trained animals. However, we found it difficult to apply the same threshold to analyze all of the trained animals, because the trained animals often displayed calcium transients that were irregular in time and generated time-derivatives that fluctuated between positive values and negative values (Fig. 1e and 1f). Therefore, instead of using the amplitude of time-derivative as a threshold, we used the amplitude of calcium response as a threshold to measure response duration. We used three different thresholds (10%, 30% and 50% change of the baseline

calcium signal) and obtained the same conclusion (Fig. 2b, 2e, 3e, 5f, 5i, 6d and Supplementary Fig. 5 and 6). Thus, we hope that our newly added analyses (cross-correlations with a sliding window, state analysis over multiple time windows) together with our original analyses have addressed the reviewer's concerns.

The results in figure 3h-k are really nice.
The results in figure 6 are nice.

We appreciate the reviewer's comments.

Figure 7: Why is the GFP::INX-4 protein regulated differently than the GFP protein? The authors claim it's likely not due to differences in transcription, so I'm assuming it's either translational or degradation regulation? They quantify fluorescence in the soma, but wouldn't the native signal be in the process? Are they quantifying differences in ER-localized innexin, or is their construct mislocalized? Was any difference in expression observed in the processes where the innexin should be localized?

Our results showed that training did not alter the expression of the *Pinx-4::GFP* transcriptional fusion, but decreased the level of the *Pinx-4::GFP::inx-4* translational fusion in RIM, suggesting that aversive training alters GFP::INX-4 through post-transcriptional mechanisms. We have discussed potential mechanisms in Discussion section. We agree with the reviewer that we should be able to find GFP::INX-4 at the electrical synapses on RIM processes. However, because *inx-4* is expressed in multiple neurons that extend their processes into the nerve ring, we could not identify the signal of GFP::INX-4 in the processes of RIM. We agree with the reviewer that the GFP::INX-4 signal in RIM cell body is a proxy for the level of GFP::INX-4 in the process and likely includes the protein that is being made on ER or is newly made before localized to the synapse. We have clarified this point in Discussion.

In addition, we confirmed that the GFP::INX-4 fusion expressed by the *Pinx-4::GFP::inx-4* transgene was functional, because it rescued the hypersensitive response to quinine in the *inx-4* mutant animals. Previously, it was shown that the *inx-4* mutant animals were hypersensitive in response to the repulsive chemical quinine (Krzyzanowski et al., PLoS Genetics 2016). This phenotype was rescued by expressing *Pinx-4::GFP::inx-4* (Supplementary Fig. 13). Thus, the training-dependent regulation of GFP::INX-4 in RIM likely represents the regulation of INX-4 in RIM.

It seems like the ASH/AWC story is parallel to the AIB-RIM-AVA story, but there appears to be a lack of connection between the two. How does the increase of PA14-induced activity in ASH lead to the activation of NMR-1 in RIM? Is it direct activation of RIM by ASH? Presumably ASH is still getting activated in the Cx36 background, but the AWC signal is dominating in this context? I think the ASH/AWC story is interesting, but it doesn't seem to directly connect to the AIB-RIM-AVA story which is investigated more thoroughly. Also, with regard to AWC, it appears that while the hyperpolarized state of AWC in the trained background is unaltered, it seems like the rate of decay is slower in the trained context.

Overall, I think this is a good study, but I think some re-analysis would help communicate the results better, and I think the ASH/AWC story should be dropped, or how it relates to the AIB-RIM-AVA story should be fleshed out some more.

We appreciate the reviewer's comments. Our findings showed that aversive training decreased the level of INX-4 in RIM. However, only the calcium signals evoked by PA14, but not by OP50, became decoupled after training, suggesting that PA14 evokes specific sensory responses in trained animals. Our findings showing that ASH is activated by PA14, but not by OP50, in trained animals identified this specific sensory

response. While ASH plays a role in sensing PA14 in trained animals, INX-4 downregulation induces persistent circuit changes which allow the activities of RIM, AIB and AVA to decouple in response to the inputs from ASH and AWC. These findings together with the NMR-1-UNC-43 story lead us to propose the model in Supplementary Fig. 14. It shows that in trained animals activating ASH by PA14 provides a repulsive sensory input, which is conflicting with the attractive input from AWC, to the RIM-AIB-AVA circuit that has weakened gap junctions between RIM and AIB, resulting in the decorrelation of the RIM-circuit activities. Our results show that training modulates the ASH response to PA14 and modulates the gap junction between RIM and AIB, both of which are important for learning.

We agree with the reviewer that it will be informative to further investigate the relationship between training-dependent regulation of ASH and training-dependent regulation of the gap junctions. In our original manuscript, we showed that ablating ASH disrupted learning. In the revised manuscript, we asked whether ablating ASH also disrupted training-dependent downregulation of INX-4 in RIM. We showed that in the animals that ASH was ablated, the level of GFP::INX-4 in RIM continued to be downregulated by training. These results indicate that training-dependent regulation of ASH and training-dependent regulation of RIM gap junctions likely use independent mechanisms. We included these results in Fig. 7 and discussed these findings in the Discussion section.

With regard to the rate of decrease in PA14-evoked calcium signals in AWC, we analyzed the time derivatives of the changes of the calcium signals for the 0 - 5 second, 0 - 10 second, 0 - 15 second and 0 - 30 second time windows after the switch to PA14. There is no significant difference in any of the time windows (see below).

Reviewer #3 (Remarks to the Author):

In this elegant study, the authors dissect how olfactory learning is controlled by the modulation of gap junction expression at single neuron resolution. Using neuron-specific calcium imaging, behavioral analysis and gap junction expression measurements they show that the experience of learning dampens gap junction protein expression by a conserved regulatory pathway. As such, their study reveals how regulation of gap junction protein levels can lead to experience-dependent changes in behaviour. The research findings are novel and will be of interest to the wider field. The manuscript is very well written and the extensively controlled data are meticulously presented and fully support their conclusions. However, the following areas need to be made more clear:

We appreciate the reviewer's comments.

Line 181-183 - the rationale leading to this hypothesis below is not strong. Please revise. "We hypothesized that the training-induced changes in PA14-evoked activities of AIB, AVA and RIM, such as the decoupling of the activities, resulted from weakening of the gap junctions."

We agree with the reviewer and have revised the rationale as the following.

Revised version:

“To characterize the function of training-induced activity changes in AIB, AVA and RIM, we sought the underlying molecular and cellular mechanisms. AIB, AVA and RIM are connected through chemical synapses and gap junction - mediated electrical synapses (Fig. 3a). Because gap junctions play a critical role in coupling neuronal activities, we examined whether the training-dependent changes in AIB, AVA and RIM, such as the decoupling of the neuronal activities, resulted from weakening of the gap junctions expressed in these cells. We expressed the primary mammalian neuronal gap junction molecule connexin Cx36 (Fig. 3a) using promoters selectively expressed in these neurons.”

Line 222-223 - The sentence containing this (“do not change their response to PA14 odorants”) is not clear. This sentence reads that the AWCs do not respond to PA14 odorants at all but the calcium imaging clearly shows that PA14 odor suppresses AWC activity. Therefore, the AWCs do change their response to PA14 but not to training.

We apologize for the confusion and have revised the sentence.

Revised version:

“In comparison, the main olfactory sensory neurons AWC that sense food odorants upstream of AIB, AVA and RIM do not show a significant change in their PA14 - evoked calcium responses after training (Fig. 4e and 4f) and continue to respond to PA14 as an attractive cue.”

Line 224 - There seems to be some residual learning upon ASH ablation. Perhaps the term “completely abolished” needs to be revised in the followings sentence?
“Genetically ablating ASH completely abolished the aversive learning of PA14 (Fig 4g).

We agree with the reviewer and have revised the sentence as “Genetically ablating ASH significantly disrupted aversive learning of PA14”.

Line 232 - the rationale behind examining the role of NMR-1 is weak. The authors must introduce this section with more clarity.

We have revised the rationale as the following:

“Next, we sought the mechanisms underlying training-induced decoupling. We examined *nmr-1*, because it encodes the *C. elegans* homolog of the NR1 subunit of the mammalian NMDA-type glutamate receptors (NMDARs), which play a critical role in regulating synaptic plasticity in the mammalian brain. Meanwhile, *nmr-1* is expressed in a few *C. elegans* neurons, including RIM.”

I also identified a few typographical that need to be amended:

Line 58 - remove “the” after “detect”

Line 73 - change “show” to “showed” after previously

Line 79 - remove “the” before “training-dependent” and before “RIM gap”

Line 154 - change “totally” to “in total” before “eight”

Line 297 - change “somata” to “soma”

Line 357 - change “systems” to “system”

We appreciate the reviewer’s help and have corrected these mistakes.

The greek omega sign should be defined the first time it is mentioned.

We have now defined the omega bend the first time it is mentioned.

Reviewers' Comments:

Reviewer #1:

Remarks to the Author:

Excellent paper. The authors have added additional experiments that have addressed the concerns raised.

Reviewer #2:

Remarks to the Author:

I am satisfied with the edits and additional analysis performed by the authors, and endorse this manuscript for publication. This was an enjoyable read.

Reviewer #3:

Remarks to the Author:

My initial minor concerns have been fully addressed in review.

Response to the reviewers' comments.

REVIEWERS' COMMENTS:

Reviewer #1 (Remarks to the Author):

Excellent paper. The authors have added additional experiments that have addressed the concerns raised.

Reviewer #2 (Remarks to the Author):

I am satisfied with the edits and additional analysis performed by the authors, and endorse this manuscript for publication. This was an enjoyable read.

Reviewer #3 (Remarks to the Author):

My initial minor concerns have been fully addressed in review.

We appreciate the reviewers' comments.